# PID-Guided Partial Alignment for Multimodal Decentralized Federated Learning

## Abstract

Multimodal decentralized federated learning (DFL) is challenging because agents differ in available modalities and model architectures, yet must collaborate over peer-to-peer (P2P) networks without a central coordinator. Standard multimodal pipelines learn a single shared embedding across all modalities. In DFL, a monolithic representation induces gradient misalignment between uni- and multimodal agents; as a result, it suppresses heterogeneous sharing and cross-modal interaction. We present PARSE, a multimodal DFL framework that *operationalizes* partial information decomposition (PID) in a server-free setting. Each agent performs *feature fission* to factorize its latent into *redundant*, *unique*, and *synergistic* slices. P2P knowledge sharing among heterogeneous agents are enabled by slice-level *partial alignment*: only semantically shareable branches are exchanged among agents that possess the corresponding modality. By removing the need for central coordination and gradient surgery, PARSE resolves uni–/multimodal gradient conflicts, thereby overcoming the multimodal DFL dilemma while remaining compatible with standard DFL constraints. Across benchmarks and agent mixes, PARSE yields consistent gains over task-, modality-, and hybrid-sharing DFL baselines. Ablations on fusion operators and split ratios, together with qualitative visualizations, demonstrate the efficiency and robustness of the proposed design.

## 1 Introduction

Decentralized federated learning (DFL) (Yuan et al., 2024b) enables agents to train collaboratively over peer-to-peer (P2P) networks without a central coordinator, improving robustness and privacy relative to server-based FL (McMahan et al., 2017). In realistic deployments such as autonomous systems (Caesar et al., 2020; He et al., 2021; Zheng et al., 2023; Cui et al., 2024), healthcare (Frantzidis et al., 2010; Zhang et al., 2020; Bertsimas & Ma, 2024), and human-computer interaction (Gao et al., 2020; Liu et al., 2021; Lv et al., 2022; Moin et al., 2023), agents rarely have the same set of sensors or models; instead, *modality heterogeneity* is the norm. This makes *multimodal DFL* especially challenging: agents own different modality subsets (architecture mismatch), exchange updates over sparse graphs without a server to arbitrate or re-balance updates (limited consensus), and often exhibit conflicting gradients between uni- and multi-modal agents.

Traditional multimodal pipelines learn a single shared embedding across heterogeneous agents. While effective in centralized training, such *monolithic* representations are brittle in DFL because: (i) agents with different modality sets push incompatible updates to the same parameters, inducing gradient conflicts; (ii) these conflicts on shared parameters erode both cross-modal interaction and inter-agent sharing (Ouyang et al., 2023); and (iii) without a coordinator, achieving and maintaining alignment over a decentralized graph is intrinsically difficult. Section 2 shows that existing multimodal DFL strategies mirror these tensions: task-based sharing (Xiong et al., 2022) aligns only agents with identical modality sets and misses transfer across partially overlapping sets; modality-based sharing (Yuan et al., 2024a) preserves per-modality learning but forfeits cross-modal interaction; and hybrid schemes (Chen & Li, 2022) exploits interaction benefits but suffer from gradient conflicts. Motivated by these challenges, the primary question we seek to address is:

> *How can we design multimodal knowledge representation and sharing mechanisms that maximally facilitate knowledge transfer among heterogeneous peers with diverse modalities, exploit cross-modal interactions while avoiding gradient conflicts in a DFL setting?*

Table 1: Comparison with representative multimodal centralized and decentralized FL methods.

| Method | Server-free | Topology-agnostic | Gradient surgery-free | Synergistic |
|---|---|---|---|---|
| FedMSplit (Chen & Zhang, 2022) | ✗ | ✗ | ✗ | ✗ |
| Harmony (Ouyang et al., 2023) | ✗ | ✗ | ✓ | ✗ |
| DMML-KD (Yin et al., 2024) | ✗ | ✗ | ✓ | ✗ |
| MCARN (Yang et al., 2024) | ✗ | ✗ | ✓ | ✗ |
| FedHKD (Wang et al., 2024) | ✗ | ✗ | ✗ | ✗ |
| FedMVD (Gao et al., 2025) | ✗ | ✗ | ✓ | ✗ |
| PARSE (ours) | ✓ | ✓ | ✓ | ✓ |

Guided by partial information decomposition (PID), which disentangles learnable knowledge of multimodal training into *redundant* (shared across modalities), *unique* (modality-specific), and *synergistic* (emerging from modality interactions) information (Williams & Beer, 2010; Bertschinger et al., 2014; Liang et al., 2023), we propose PARSE, a multimodal DFL framework with two key designs: **(i) Feature Fission:** each agent's encoder produces a latent that is explicitly factorized into three slices, namely redundant ($z^r$), unique ($z^u$), and synergistic ($z^s$). Each agent only updates the slices corresponding to its available modalities, preventing interference across unrelated parts of the model. **(ii) P2P Knowledge Sharing with Partial Alignment:** A pair of agents perform *slice-level knowledge sharing* if and only if they share align-able slices, maximizing sharing opportunities among heterogeneous agents while avoiding gradient conflicts on slices that cannot be aligned.

As summarized in Table 1, compared to representative multimodal centralized and decentralized FL methods, PARSE offers the following properties, which constitute our main contributions:

- **Server-free.** Unlike Harmony (Ouyang et al., 2023), FedMVD (Gao et al., 2025) and MCARN (Yang et al., 2024), which require a centralized controller, PARSE is *server-free*: slice-level alignment emerges automatically by *sharing only modality-specific branches* between heterogeneous agents sharing that modality, requiring no global server orchestration.

- **Topology-agnostic.** PARSE operates over *arbitrary* (even time-varying) P2P overlays: modality-conditioned consensus runs on per-modality subgraphs, making it *topology-agnostic* and compatible with fixed communication graphs or time-varying random graphs.

- **Gradient surgery-free.** Centralized approaches such as FedHKD (Wang et al., 2024) and FedM-Split (Chen & Zhang, 2022) mitigate gradient conflicting via gradient regularization, decomposition, or other "surgeries". PARSE prevents gradient collisions without costly gradient manipulation.

- **Synergy-aware.** Decomposition-based methods such as DMML-KD (Yin et al., 2024) and FedHKD (Wang et al., 2024) disentangle redundant and unique parts but ignore *synergistic* information learnable only by multimodal agents. PARSE explicitly models this additional synergistic component, thereby yielding accuracy gains over decomposition-only baselines.

## 2 MOTIVATION STUDY: REPRESENTATION ALIGNMENT AND SHARING

Multimodal DFL presents two central challenges: *(1) how to effectively represent information across heterogeneous modalities, and (2) how to enable knowledge sharing among agents that observe different subsets of modalities*. These challenges are tightly coupled, and their joint resolution is critical for ensuring efficient and stable training. Learning a single shared embedding across all modalities, while enforcing knowledge sharing among local neighbors with differing modality sets, is prone to failure in server-free DFL settings. This is primarily due to the absence of a global coordination mechanism to mitigate gradient conflicts and training drifts caused by modality heterogeneity across agents.

### 2.1 LIMITATIONS OF EXISTING METHODS WITH FULL REPRESENTATION SHARING

A logical approach to addressing multimodal gradient conflicts is *knowledge sharing restricted to fully aligned representations*. We compare three typical designs following this approach in Fig. 1: **Task-based** (Xiong et al., 2022): all agents with the same subset of modalities form an overlay to learn a joint representation for the available modalities, train a fused model end-to-end; since all agents in the overlay share the same modality subset, the learned joint representation is fully aligned and captures the interaction between the available modalities. However, it does not facilitate

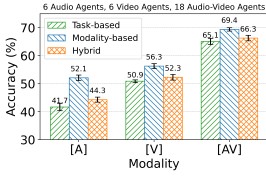 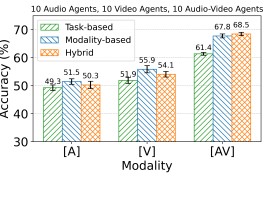 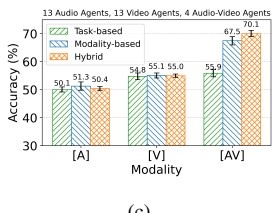

(a)           (b)           (c)

Figure 2: Test accuracy for unimodal and multimodal agents under task-, modality-, and hybrid sharing across three agent mixes: 6 audio, 6 video, and 18 multimodal agents; (b) 10 audio, 10 video, and 10 multimodal agents; and (c) 13 audio, 13 video, and 4 multimodal agents. [A], [V], [AV] are averages over audio-only, video-only, and multimodal agents.

knowledge sharing between agents with different modality subsets; **Modality-based** (Yuan et al., 2024a): all agents with a specific modality form an overlay to learn a common modality-specific representation, which is fully aligned within the modality, but does not capture interaction with the other modalities. **Hybrid** (Chen & Li, 2022): modality-based sharing plus local fusion on multimodal agents; while each modality-specific representation is still fully shared among agents, local fusion on multimodal agents diverges their training objectives from unimodal agents.

We evaluate the performance of the above three designs using DSGD (Lian et al., 2017) on AVE (Tian et al., 2018) with 30 agents and three modality compositions (Fig. 2) (see implementation details in Appendix B).

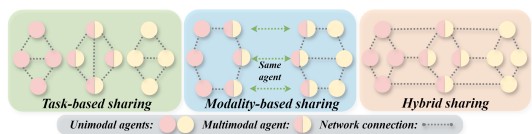

Figure 1: We study three knowledge-sharing strategies in DFL setting. To illustrate, we consider a two-modality scenario involving three types of agents: *Red:* modality A only; *yellow:* modality B only; *bicolored:* both.

• *Task-based sharing has the worst performance.* While agents with identical modality subset are fully aligned, this silo-style training prevents knowledge sharing between agents with partially overlapping modality subsets, seriously limiting learning efficiency of DFL.

• *Modality-based sharing achieves strong unimodal performance, but weak multimodal performance.* Training and aggregating on a modality-specific overlay maximizes within-modality knowledge sharing and fully aligns local objectives. However, lacking cross-modal interaction undermines multimodal performance, which trails the hybrid scheme, and the gap widens as the unimodal-agent ratio increases.

• *Hybrid sharing fails to fully exploit multimodal interaction.* It achieves only marginal gains for *unimodal* agents over task-based sharing, with modest *multimodal* improvements as the unimodal ratio increases (Fig. 2 (b),(c)). This pattern confirms *knowledge misalignment*: because agents with different modality subsets access different information, their gradients pull shared parameters towards different directions. On closer inspection, Fig. 3 shows *strong alignment* between agents with the same subset of modalities, and *weak alignment* between agents with different modality subsets when updating the same modality branch.

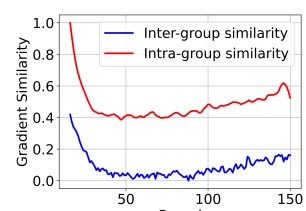

Figure 3: Group agents by their modality subsets, we show inter-group gradient cosine similarity (w.r.t. shared models) and intra-group gradient cosine similarity.

**Key Takeaways.** *1) To maximize DFL learning efficiency with limited neighborhood connectivity, it is crucial to facilitate P2P knowledge sharing between heterogeneous agents that possess different subsets of data modalities; 2) Without a global coordination mechanism, enforcing a unified, modality-specific representation across heterogeneous agents leads to knowledge misalignment, limiting the effectiveness and stability of multimodal DFL.*

## 2.2 CROSS-MODALITY SHARING BASED ON FEATURE FISSION AND PARTIAL ALIGNMENT

Deviating from the monolithic design, we propose to decompose each modality-specific representation into multiple disentangled subspaces, or slices, each encoding distinct facets of multimodal

information (*feature fission*). To mitigate inter-agent gradient conflicts, we introduce *partial alignment*, a selective sharing mechanism that restricts representation exchange to only those slices that are mutually alignable, thereby promoting stable and effective decentralized multimodal learning.

**Feature Fission through Partial Information Decomposition.** For two modalities $X_1$ and $X_2$ predicting $Y$, partial information decomposition (PID) (Williams & Beer, 2010; Bertschinger et al., 2014; Liang et al., 2023) separates

$$I(X_1, X_2; Y) \;=\; \underbrace{U_{X_1}(Y) + U_{X_2}(Y)}_{\text{unique}} \;+\; \underbrace{R(Y)}_{\text{redundant}} \;+\; \underbrace{S(Y)}_{\text{synergistic}} \,, \tag{1}$$

where $R(Y)$ is information about $Y$ present in *both* modalities, $U_{X_i}(Y)$ is *only* in $X_i$, and $S(Y)$ arises *only* under joint observation. We illustrate PID with two modalities for clarity, but both PID and our design extend naturally to more than two modalities (Griffith & Koch, 2014).

Motivate by PID, the representation for a modality can be decomposed into three slices—*redundant* $(R(Y))$, *unique* $(U_{X_i}(Y))$, and *synergistic* $(S(Y))$. PID provides clear principles on how the slices *should* be aligned and shared across agents: (i) *unique* slice is exclusive to a modality and is shareable among all agents that possess that modality; (ii) *redundant* slice is shareable across all agents, though harder for unimodal agents to estimate; (iii) *synergistic* slice captures cross-modal complementarity, is learnable only by multimodal agents (still heterogeneous across different modality subsets).

**P2P Knowledge Sharing with Partial Alignment.** With slicing, each agent only updates the slices corresponding to its available modalities, preventing interference across unrelated parts of the model. *Slice-level alignment* emerges automatically by sharing modality-specific branches between heterogeneous agents. Knowledge sharing between a pair of agents is restricted to their *align-able* slices, therefore not distracted by gradient conflicts between slices that cannot be aligned. P2P knowledge sharing is compatible with any DFL communication topology and does not require any costly coordination or gradient surgery. These considerations motivate the `PARSE` framework introduced next (Section 3).

# 3 THE `PARSE` FRAMEWORK FOR MULTIMODAL DFL

Motivated by Section 2, we formally present `PARSE`, a server-free multimodal DFL framework for heterogeneous agents with arbitrary modality mixes. Fig. 4 illustrates the two-modality, three-agent case for clarity, while our framework supports any number of modalities and agents. `PARSE` couples an encoder-classifier architecture with *feature fission* and *partial alignment*: each modality $m$ has a feature encoder $h^m$ whose output latent is factorized into redundant $(z^r)$, unique $(z^u)$, and synergistic $(z^s)$ slices; unimodal agents train unique and redundant heads on their local data, while multimodal agents additionally fuse $z^s$ to train a synergistic head. This design lets agents exchange what should be shared while preserving modality-exclusive capacity and learning synergy without a server.

## 3.1 PROBLEM SETUP

Let $\mathcal{M}$ denote the global set of modalities. We consider a set of agents $\mathcal{N}$, where agent $i \in \mathcal{N}$ owns a subset of modalities $\mathcal{M}_i \subseteq \mathcal{M}$ and a local dataset $S_{\mathcal{M}_i} = \{(\mathbf{x}_{ij}, y_{ij})\}_{j=1}^{|S_{\mathcal{M}_i}|}$ with $\mathbf{x}_{ij} = \{x_{ij}^m\}_{m \in \mathcal{M}_i}$ and $y_{ij} \in \mathcal{Y}$. All agents share the same label space $\mathcal{Y}$, while both the input distributions and the available modality subsets $\mathcal{M}_i$ may be heterogeneous and non-IID across agents.

**Model Family.** For each modality $m \in \mathcal{M}$, agents that possess $m$ use a (potentially shared) encoder $h^m : \mathcal{X}^m \to \mathcal{Z}^m$ to produce latent features.[1] For a multimodal agent $i$, an aggregator $\mathcal{A}_i : \prod_{m \in \mathcal{M}_i} \mathcal{Z}^m \to \mathcal{Z}$ maps per-modality features into a representation space $\mathcal{Z}$, and a classifier $f : \mathcal{Z} \to \mathcal{Y}$ outputs predictions. In Section 3.2 we will *instantiate* this generic family with `PARSE` by factorizing each encoder's latent into redundant/unique/synergistic slices and by using component-wise heads and partial alignment.

---

[1]Architectures can differ across modalities (e.g., CNN for vision, Transformer for text) and, when desired, be shared across agents that own the same modality.

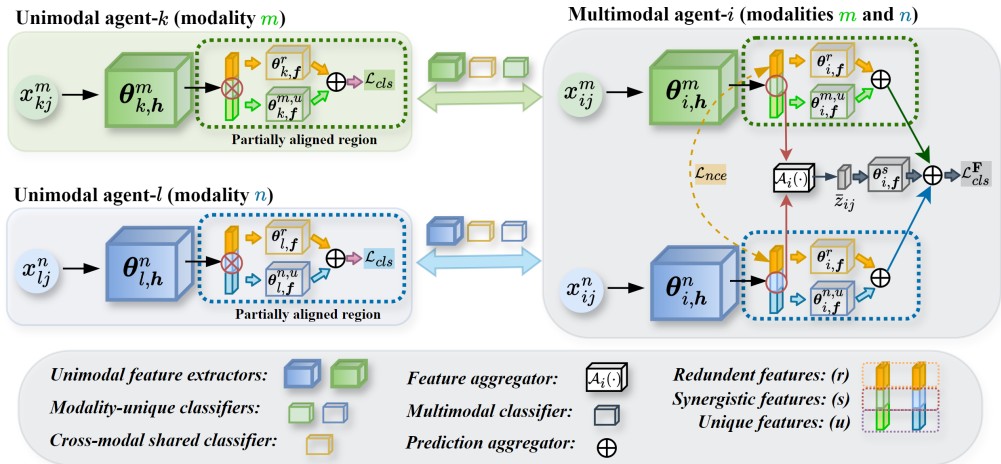

Figure 4: **PARSE at a glance (two modalities for illustration).** Agents that share a modality form a *modality-specific* P2P subgraph. Each encoder output is *fissioned* into redundant ($z^r$), synergistic ($z^s$), and unique ($z^u$) slices. Unimodal agents train on $z^u$ and $z^r$. Multimodal agents: (i) *partially align* $z^u$ and $z^r$ across agents, and (ii) learn a multimodal classifier on the fused $z^s$. This routes shareable information while keeping modality-exclusive and joint-only components local.

**Learning Objective.** Let $\boldsymbol{\theta}$ collect all trainable parameters (encoders, aggregator(s), and classifier(s)). The decentralized objective minimizes

$$\min_{\boldsymbol{\theta}} \ \frac{1}{|\mathcal{N}|} \sum_{i \in \mathcal{N}} \mathbb{E}_{(\mathbf{x},y) \sim S_{\mathcal{M}_i}} \left[ \mathcal{L}_i(\mathbf{x}, y; \boldsymbol{\theta}) \right], \quad (2)$$

the average local loss in Eq. (2), where $\mathcal{L}_i$ is a task-appropriate loss (e.g., cross-entropy). Due to modality heterogeneity, agent $i$ updates only the parameters tied to $\mathcal{M}_i$ (its encoders for $m \in \mathcal{M}_i$ and any heads it hosts).

**Decentralized P2P Communication.** Agents exchange parameters with graph neighbors in a P2P fashion. To respect modality availability, communication occurs on per-modality subgraphs among agents that own the same modality. the precise update rule (DSGD with mixing matrices) and the set of parameters exchanged are detailed in Section 3.4.

## 3.2 FEATURE FISSION

To separate what is shareable from what must remain local (as motivated in Section 2), each encoder output is factorized into PID-motivated slices. For $x_{ij}^m$,

$$z_{ij}^m = h^m(x_{ij}^m; \boldsymbol{\theta}_{i,\boldsymbol{h}}^m) = \text{Concat}\left(z_{ij}^{m,r}, z_{ij}^{m,s}, z_{ij}^{m,u}\right), \quad (3)$$

where $z^{m,r}$ captures *redundant* information (present in all modalities), $z^{m,s}$ captures *synergy* (available only under joint observation), and $z^{m,u}$ captures *unique* modality-specific cues. We adopt an equal split by default, and Section 4 reports split-ratio sweeps.

This fission supplies a routing mechanism in DFL: neighbors exchange modality-specific parameters, with shareable information automatically aligned ($z^u$ and $z^r$).

## 3.3 LOSSES AND PARTIAL ALIGNMENT

**Modality-unique and redundant Training.** Each agent trains two heads per available modality $m \in \mathcal{M}_i$: a unique head $f_i^u(\cdot; \boldsymbol{\theta}_{i,\boldsymbol{f}}^{m,u})$ on $z^{m,u}$ and a redundant head $f_i^r(\cdot; \boldsymbol{\theta}_{i,\boldsymbol{f}}^r)$ on $z^{m,r}$. The per-modality prediction is $\hat{y}_{ij}^m = f_i^u(z_{ij}^{m,u}) + f_i^r(z_{ij}^{m,r})$, with classification loss

$$\mathcal{L}_{cls}(\boldsymbol{\theta}_{i,\boldsymbol{h}}^m, \boldsymbol{\theta}_{i,\boldsymbol{f}}^{m,u}, \boldsymbol{\theta}_{i,\boldsymbol{f}}^r) = \frac{1}{|S_{\mathcal{M}_i}|} \sum_j \ell\left(\hat{y}_{ij}^m, y_{ij}\right), \quad (4)$$

where $\ell : \mathcal{Y} \times \mathcal{Y} \to \mathbb{R}$ is a task-appropriate loss function (e.g., cross-entropy).

**Contrastive Diversity for redundant Alignment.** Only redundant slices are aligned across modalities of the *same* sample, while unique/synergistic slices act as hard negatives to encourage orthogo-

nality. To this end, we impose a contrastive objective $\mathcal{L}_{nce}$ on the redundant features:

$$\mathcal{L}_{nce}(\Theta_{i,\boldsymbol{h}}) = \frac{-1}{|S_{\mathcal{M}_i}|} \sum_{j} \sum_{\substack{(m,m') \in \mathcal{M}_i \\ m \neq m'}} \log \frac{\exp\big(\mathrm{sim}(z_{ij}^{m,r}, z_{ij}^{m',r})/\tau\big)}{\exp\big(\mathrm{sim}(z_{ij}^{m,r}, z_{ij}^{m,u})/\tau\big) + \exp\big(\mathrm{sim}(z_{ij}^{m,r}, z_{ij}^{m,s})/\tau\big)}, \quad (5)$$

where $\Theta_{i,\boldsymbol{h}} = \{\boldsymbol{\theta}_{i,\boldsymbol{h}}^m\}_{m \in \mathcal{M}_i}$, $\mathrm{sim}(\cdot,\cdot)$ denotes a similarity metric (e.g., cosine similarity), and $\tau$ is a temperature hyperparameter.

**Synergy on Multimodal Agents.** For multimodal agent $i$, synergistic features are fused by a parameter-free mean $\bar{z}_{ij} = \mathcal{A}_i(\{z_{ij}^{m,s}\}_{m \in \mathcal{M}_i}) = \frac{1}{|\mathcal{M}_i|} \sum_{m \in \mathcal{M}_i} z_{ij}^{m,s}$, then classified by $f_i^s(\cdot; \boldsymbol{\theta}_{i,\boldsymbol{f}}^s)$ with an ensemble loss that encourages cooperation with per-modality predictions:

$$\mathcal{L}_{cls}^{\boldsymbol{F}}(\Theta_{i,\boldsymbol{h}}, \Theta_{i,\boldsymbol{f}}^u, \boldsymbol{\theta}_{i,\boldsymbol{f}}^r, \boldsymbol{\theta}_{i,\boldsymbol{f}}^s) = \frac{1}{|S_{\mathcal{M}_i}|} \sum_{j} \ell\left(f_i^s(\bar{z}_{ij}) + \sum_{m \in \mathcal{M}_i} \hat{y}_{ij}^m, y_{ij}\right), \quad (6)$$

with $\Theta_{i,\boldsymbol{f}}^u = \{\boldsymbol{\theta}_{i,\boldsymbol{f}}^{m,u}\}_{m \in \mathcal{M}_i}$ being the collection of all unimodal classifier parameters.

**Local Objectives.** Unimodal agent $i$ minimizes $\mathcal{L}_{cls}$, while multimodal agent $i$ minimizes

$$\mathcal{L}_i = \mathcal{L}_{cls}^{\boldsymbol{F}}(\cdot) + \beta \cdot \mathcal{L}_{nce}(\cdot), \quad (7)$$

where $\beta$ balances alignment. Inference uses the ensemble $\hat{y}_{ij} = f_i^s(\bar{z}_{ij}) + \sum_{m \in \mathcal{M}_i} \hat{y}_{ij}^m$.

**Partial Alignment.** Rather than training the three branches in isolation, we optimize an *ensemble* prediction (Eq. (6)) in which each modality contributes a unique head $f^u$ and a shared redundant head $f^r$, and multimodal agents add a synergistic head $f^s$. The contrastive-diversity loss in Eq. (5) (i) *pulls* redundant features across modalities for the *same* sample (positives), and (ii) treats the sample's own unique and synergistic components as *hard negatives*, thereby encouraging $z^r$, $z^u$, and $z^s$ to capture complementary information. Crucially, only the redundant slice $z^r$ (and its head $f^r$) participates in cross-agent synchronization; $z^u$ and $z^s$ are explicitly pushed away from $z^r$ by Eq. (5) and are optimized *locally*. This "partial" alignment mitigates uni- vs. multi-modal gradient conflict, preserves modality-exclusive capacity, and lets synergy be learned where it exists (on multimodal agents) without contaminating unimodal updates. In practice, the ensemble objective allows agents with different modality sets to share a common optimization signal through the aligned redundant branch while keeping non-shared branches non-interfering, yielding stable improvements even if any single sub-predictor is imperfect (cf. Eq. (7)).

## 3.4 KNOWLEDGE SHARING

We adopt decentralized SGD (DSGD) (Lian et al., 2017), where agents exchange parameters only with graph neighbors. To respect modality heterogeneity, each multimodal agent is instantiated as multiple *unimodal virtual agents*, one per owned modality, and each virtual agent participates in a *modality-specific* communication overlay.

**Per-modality Subgraphs.** For each modality $m \in \mathcal{M}$, we construct a subgraph $\mathcal{G}^m = (\mathcal{N}^m, \mathcal{E}^m)$ with $\mathcal{N}^m = \{i \in \mathcal{N} \mid m \in \mathcal{M}_i\}$. An edge $(i, k) \in \mathcal{E}^m$ means agents $i$ and $k$ can exchange updates for modality $m$. Each subgraph is equipped with a mixing matrix $\mathbf{W}^m \in \mathbb{R}^{|\mathcal{N}^m| \times |\mathcal{N}^m|}$ (row-stochastic, respecting the sparsity of $\mathcal{E}^m$), where $W_{ik}^m$ is the weight agent $i$ assigns to neighbor $k$ on modality $m$.

**Neighbor Mixing.** At each communication round, agent $i$ performs a local step and then mixes with its neighbors:

$$\boldsymbol{\theta}_i^m \leftarrow \sum_{k \in \mathcal{N}^m} W_{ik}^m \big(\boldsymbol{\theta}_k^m - \eta \nabla_{\boldsymbol{\theta}_k^m} \mathcal{L}_k\big), \quad (8)$$

where $\eta$ is the learning rate, and $\boldsymbol{\theta}_i^m = \{\boldsymbol{\theta}_{i,\boldsymbol{h}}^m, \boldsymbol{\theta}_{i,\boldsymbol{f}}^{m,u}, \boldsymbol{\theta}_{i,\boldsymbol{f}}^r\}$ collects the modality-$m$ encoder, the modality-unique head, and the shared redundant head maintained by agent $i$. Subgraphs $\mathcal{G}^m$ can use ring, exponential, or other sparse topologies without changing the update rule.

**Synergistic-head Subgraphs.** Parameters of the multimodal (synergistic) classifier $\boldsymbol{\theta}_{i,\boldsymbol{f}}^s$ are exchanged only among agents that share the same modality set $\mathcal{M}_i$ (a small subgraph per set). Since $f_i^s$ is a single linear layer, this additional message is negligible compared with exchanging encoders.

***Summary.*** By communicating on per-modality subgraphs and mixing only the parameters tied to owned modalities, PARSE routes shareable information across peers while preserving the integrity of unshared parts, enabling stable collaboration without a server.

| Agent ratios | | 6 : 6 : 18 | | | 10 : 10 : 10 | | | 13 : 13 : 4 | | |
|---|---|---|---|---|---|---|---|---|---|---|
| **Agent types** | | [A] | [G] | [AG] | [A] | [G] | [AG] | [A] | [G] | [AG] |
| **KU-HAR** | DSGD-Modality | 80.1±0.7 | 68.4±1.4 | 85.2±1.7 | 77.3±0.9 | 68.1±0.5 | 83.9±1.0 | 78.6±1.3 | 71.9±1.6 | 83.1±1.5 |
| | DSGD-Task | 74.8±0.6 | 64.4±2.3 | 83.2±0.6 | 77.4±0.4 | 67.4±1.2 | 80.2±0.5 | 76.9±0.5 | 70.7±1.4 | 77.8±0.9 |
| | DSGD-Hybrid | 74.3±0.8 | 62.9±1.4 | 84.4±1.3 | 75.3±0.7 | 64.9±1.2 | 85.3±0.4 | 77.3±0.3 | 70.6±0.5 | 86.0±1.2 |
| | Harmony | 78.9±0.6 | 67.3±0.7 | 87.4±0.6 | 77.8±1.2 | 67.7±1.2 | 87.5±0.8 | 77.6±0.6 | 71.5±0.7 | 88.1±0.6 |
| | FedHGB | 76.3±0.8 | 66.8±1.0 | 82.6±0.8 | 76.1±0.7 | 65.6±0.4 | 84.5±0.7 | 76.5±0.5 | 71.1±0.7 | 84.6±0.5 |
| | DMML-KD | 78.3±0.4 | 63.0±1.1 | 86.5±0.6 | 76.9±0.6 | 66.1±0.5 | 88.0±0.9 | 79.0±0.7 | 70.4±0.5 | 87.2±0.6 |
| | PARSE | 80.6±1.0 | 68.4±1.1 | 88.1±0.4 | 79.4±0.2 | 68.4±0.2 | 88.6±0.5 | 80.9±0.4 | 73.1±0.7 | 88.4±0.5 |
| **Agent types** | | [V1] | [V2] | [V1,V2] | [V1] | [V2] | [V1,V2] | [V1] | [V2] | [V1,V2] |
| **ModelNet-40** | DSGD-Modality | 77.8±0.6 | 71.6±1.5 | 75.7±0.8 | 72.4±2.0 | 67.9±1.8 | 73.4±1.0 | 75.2±2.0 | 69.1±0.9 | 76.4±0.7 |
| | DSGD-Task | 72.3±1.2 | 67.1±0.9 | 73.3±1.3 | 73.0±1.8 | 65.6±0.8 | 71.4±0.9 | 73.2±0.7 | 68.1±0.7 | 66.2±1.4 |
| | DSGD-Hybrid | 72.1±0.5 | 59.8±1.1 | 72.9±1.4 | 72.3±1.2 | 65.8±0.7 | 72.2±1.1 | 69.4±0.7 | 67.7±1.0 | 73.3±1.2 |
| | Harmony | 74.1±1.3 | 71.5±1.0 | 77.3±0.4 | 76.4±2.0 | 67.8±1.5 | 77.2±2.3 | 75.1±1.2 | 69.1±1.6 | 76.7±0.5 |
| | FedHGB | 72.6±0.6 | 60.6±0.4 | 74.4±0.5 | 72.2±1.4 | 67.2±1.0 | 74.1±1.6 | 69.3±1.7 | 65.1±2.0 | 74.9±0.9 |
| | DMML-KD | 76.4±1.0 | 67.6±1.3 | 75.1±1.2 | 76.4±1.4 | 66.9±2.2 | 76.4±1.6 | 72.2±1.5 | 64.3±0.5 | 78.4±0.8 |
| | PARSE | 78.6±1.2 | 71.9±0.6 | 78.8±0.6 | 77.9±0.4 | 69.8±1.2 | 79.3±0.9 | 75.8±1.0 | 70.9±1.2 | 81.2±0.7 |
| **Agent types** | | [A] | [V] | [AV] | [A] | [V] | [AV] | [A] | [V] | [AV] |
| **AVE** | DSGD-Modality | 46.1±0.3 | 52.4±0.2 | 63.4±0.7 | 44.8±0.6 | 52.2±0.4 | 61.4±1.0 | 43.7±0.3 | 50.3±0.2 | 60.9±0.6 |
| | DSGD-Task | 35.3±0.7 | 43.3±0.5 | 58.5±0.3 | 41.0±0.7 | 49.1±0.9 | 56.6±0.8 | 42.6±0.3 | 49.6±1.0 | 50.6±1.0 |
| | DSGD-Hybrid | 37.3±0.9 | 45.7±0.8 | 60.7±0.6 | 38.1±1.4 | 50.3±0.9 | 61.0±0.7 | 41.4±0.7 | 50.1±0.4 | 62.1±0.8 |
| | Harmony | 44.1±0.5 | 49.4±1.0 | 64.6±0.6 | 42.2±0.8 | 51.7±1.1 | 64.5±0.8 | 42.6±0.7 | 50.1±1.2 | 60.3±0.7 |
| | FedHGB | 42.0±0.4 | 47.6±0.5 | 63.1±0.3 | 41.4±1.3 | 51.1±0.7 | 60.6±0.9 | 44.6±0.4 | 50.6±0.5 | 60.9±0.8 |
| | DMML-KD | 38.8±1.1 | 41.2±0.4 | 64.1±0.5 | 41.3±0.9 | 43.5±0.7 | 63.7±0.5 | 43.4±0.9 | 43.1±0.6 | 62.8±0.8 |
| | PARSE | 47.2±2.1 | 53.3±1.5 | 65.1±1.1 | 45.6±1.8 | 52.7±0.3 | 64.7±1.3 | 45.3±0.8 | 53.2±1.6 | 64.3±1.2 |

| Agent ratios | | 6 : 6 : 6 : 22 | | | | 10 : 10 : 10 : 10 | | | | 12 : 12 : 12 : 4 | | | |
|---|---|---|---|---|---|---|---|---|---|---|---|---|---|
| **Agent types** | | [A] | [V] | [T] | [AVT] | [A] | [V] | [T] | [AVT] | [A] | [V] | [T] | [AVT] |
| **IEMOCAP** | DSGD-Modality | 47.7±0.6 | 44.2±1.1 | 60.3±0.4 | 64.5±0.6 | 46.5±1.1 | 50.3±0.4 | 58.6±0.3 | 65.1±1.2 | 46.0±0.7 | 53.9±1.2 | 57.6±0.5 | 63.6±0.8 |
| | DSGD-Task | 47.2±0.3 | 44.9±0.5 | 57.2±0.8 | 68.8±0.6 | 46.0±1.2 | 50.1±0.3 | 55.4±1.1 | 68.4±0.6 | 48.1±0.3 | 53.3±0.5 | 57.1±0.2 | 64.6±0.4 |
| | DSGD-Hybrid | 45.8±0.5 | 45.5±0.3 | 60.1±0.9 | 68.3±1.1 | 46.0±0.6 | 50.2±0.5 | 58.3±0.9 | 68.3±1.3 | 48.3±0.6 | 53.8±0.4 | 57.4±0.8 | 68.4±0.7 |
| | Harmony | 47.6±1.3 | 43.9±0.7 | 60.2±0.6 | 65.7±0.8 | 45.6±1.5 | 50.2±0.7 | 57.8±0.8 | 67.1±0.7 | 47.5±0.8 | 53.4±0.5 | 57.0±0.8 | 68.0±1.2 |
| | FedHGB | 46.5±1.1 | 43.3±1.6 | 60.1±0.6 | 65.6±1.1 | 46.1±0.7 | 49.7±1.0 | 57.8±0.6 | 66.7±0.8 | 47.3±0.8 | 53.2±1.1 | 56.3±0.8 | 68.2±1.2 |
| | DMML-KD | 45.2±1.1 | 43.9±1.3 | 55.7±1.3 | 71.4±0.8 | 47.9±0.7 | 48.6±1.2 | 56.7±0.5 | 71.3±1.4 | 50.2±0.5 | 50.1±1.0 | 51.6±0.6 | 71.2±0.8 |
| | PARSE | 48.2±0.6 | 47.2±0.8 | 61.4±0.2 | 73.6±0.2 | 48.9±1.2 | 50.9±1.2 | 60.3±0.2 | 73.2±0.6 | 50.5±1.3 | 54.3±1.4 | 58.8±1.0 | 74.3±0.3 |

Table 2: Performance (accuracy %) of methods under varying agent-ratio scenarios (Dirichlet $\alpha = 0.5$, ring topology). Cell shading indicates each method's relative performance within each column.

## 4 EXPERIMENTS

**Datasets.** (1) KU-HAR (Sikder & Nahid, 2021): daily activities recorded by accelerometer (A) and gyroscope (G). We follow Feng et al. (2023) and use eight classes: walking, walking upstairs, walking downstairs, sitting, standing, laying, jumping, and running. (2) ModelNet-40 (Wu et al., 2015): 40 CAD object classes; two rendered views serve as modalities {V1, V2}. (3) AVE (Tian et al., 2018): 10-s video clips from 28 events with synchronized audio and visual streams {A, V}. (4) IEMOCAP (Busso et al., 2008): audio, visual, and text modalities {A, V, T}; we use the four-class emotion setting (happy, sad, angry, neutral) from Liang et al. (2020). We simulate modality heterogeneity by assigning each agent either all available modalities or just one. Additional modality mixes and implementation details are in Appendices D and C.

**Non-IID Distribution.** We assign each agent a label distribution drawn from a Dirichlet distribution with concentration parameter $\alpha$ (Hsu et al., 2019). We simulate 30 agents (40 on IEMOCAP) and report our main results with $\alpha = 0.5$. Results for other $\alpha$ values are provided in Appendix D.

**Communication Configuration.** Following Section 2, we default to ring topologies (Koloskova et al., 2019): modality-specific rings (agents sharing a modality) or task-specific rings (agents with identical modality sets). Each round, agents train for one local epoch and exchange modality-specific parameters with their two ring neighbors.

**Comparison Methods.** Because multimodal DFL is still nascent, we adapt server-based multimodal FL methods that can operate without a central coordinator: (1) Harmony (Ouyang et al., 2023): om-stamtoated as a modality-stage followed by a task-stage; (2) FedHGB (Chen & Li, 2022): hierarchical gradient blending on a hybrid graph; and (3) DMML-KD (Yin et al., 2024): a shared feature generator on the same hybrid graph for fair comparison. We also evaluate three lightweight DSGD baselines (Section 2): (4) DSGD-Modality (Yuan et al., 2024a), (5) DSGD-Task (Xiong et al., 2022), and (6) DSGD-Hybrid (Chen & Li, 2022) (without gradient blending). For completeness, we run PARSE in a conventional server-based FL setting alongside standard multimodal FL baselines; see Appendix H.

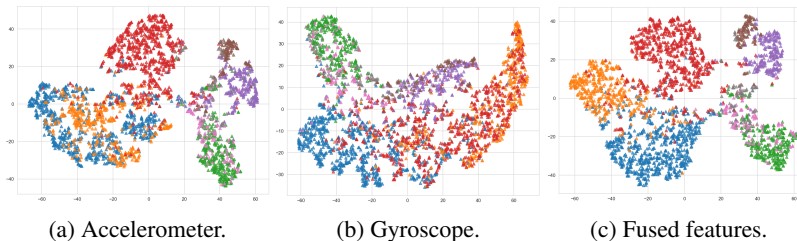

(a) Accelerometer.  (b) Gyroscope.  (c) Fused features.

Figure 5: t-SNE visualization of synergistic features from each modality and after fusion.

## 4.1 MAIN RESULTS

We evaluate all methods on four datasets and report accuracy by agent type. We denote types by their available modalities, e.g., [A] for audio-only and [AV] for audio and visual signals. Each experiment specifies an agent mix (the counts of each type). Table 2 reports the mean test accuracy for every type. The results reveal several consistent trends:

**Consistent Improvement.** On KU-HAR, `PARSE` outperforms all baselines by about 0.3–0.6% across every agent type. On ModelNet-40 and AVE, gains grow when multimodal agents are scarce, reaching 1.8–2.8% for both multimodal and the most impacted unimodal groups. On IEMOCAP, `PARSE` yields 1.9–3.1% higher accuracy on multimodal agents across all splits. Whereas existing methods tend to favor either unimodal agents (e.g., DSGD-Modality) or multimodal agents (e.g., Harmony), *PARSE provides consistent benefits to every agent type.*

**Modality Heterogeneity.** Like DMML-KD, `PARSE` disentangles redundant and unique features. In addition, it *explicitly* models synergistic subspace. As a result, it consistently attains the highest multimodal accuracy across all agent-ratio settings, while keeping unimodal performance stable as the overall modality mix varies. Concretely, Fig. 5 shows t-SNE plots of per-modality synergistic features (pre-fusion) with their fused counterpart on KU-HAR: before fusion, classes remain interleaved (e.g., blue-yellow in Fig. 5(a), red-yellow in Fig. 5(b)), whereas fusion yields well-separated clusters with larger margins. This indicates `PARSE` captures complementary cross-modal structure in synergistic slice rather than merely aligning features. Visualizations for other datasets appear in Appendix G.

Table 3: Accuracy vs. feature split size.

| Varying Split | Metric | 32d | 64d | 96d | 128d |
|---|---|---|---|---|---|
| Unique | Unique-only | 83.5 | 86.2 | **86.7** | 86.1 |
| | Combined | 87.8 | 88.6 | **88.8** | 88.5 |
| Redundant | Redundant-only | 87.2 | **87.8** | 87.4 | 87.6 |
| | Combined | **88.7** | 88.6 | 87.6 | 87.0 |
| Synergistic | Synergistic-only | 54.5 | 61.5 | 63.7 | **67.2** |
| | Combined | 86.9 | 88.6 | **89.5** | 88.3 |

Table 4: Comparison of fusion methods.

| Fusion Method | Overall (%) | Synergy (%) |
|---|---|---|
| Mean (default) | 88.6±0.5 | 61.5±1.0 |
| Concatenation + Linear | 87.1±0.9 | 60.3±0.8 |
| Summation + Linear | 89.0±1.1 | 62.2±2.1 |
| Gated Fusion | 89.0±1.4 | **64.7±1.5** |
| Cross-Attention | **89.2±1.2** | 62.1±0.7 |
| Hadamard Product | 88.6±1.3 | 58.7±1.8 |

## 4.2 ABLATION STUDY

We conduct ablations on KU-HAR. Unless stated otherwise, we use the default setting: a 10:10:10 agent ratio and a non-IID Dirichlet split with $\alpha = 0.5$. Complete results appear in Appendices E-G.

**Impact of Feature Split Ratios.** To access how split ratio affect performance, we run a split-sweep study with total feature dimension 192. In each sweep, we vary one branch's dimensionality and divide the remaining budget equally between the other two. Table 3 reports (i) the overall accuracy of multimodal agents when the branch is included in the full model and (ii) the split-only accuracy when using that branch alone. Enlarging the unique branch from $32d$ to $128d$ improves its stand-alone accuracy by about $+3$ pp, but the overall accuracy remains flat (88.5–88.8). Oversizing the redundant branch reduces overall accuracy (88.7 to 87.0). The synergistic branch helps most at a moderate size (peak overall 89.5 at $96d$). A balanced allocation across unique, redundant, and synergistic slices (our default even split) offers a robust trade-off.

**Impact of Fusion Methods.** By default, we fuse synergistic features by simple averaging. To test whether it is too crude, we compare mean fusion with five stronger operators: concatenation+Linear, summation+Linear, gated fusion (Xue & Marculescu, 2023), cross attention (Zhang et al., 2022), and Hadamard product (Kim et al., 2017), where "Linear" denotes a trainable linear layer. As reported in Table 4, gated fusion yields the highest synergistic-only accuracy, and cross-attention give a

slight overall gain over mean. However, these operators introduce extra parameters that must be exchanged and synchronized across agents each round. Mean fusion avoids this coordination cost while remaining competitive, making it a practical default for DFL.

**Impact of non-IID Degree.** We assess robustness to data heterogeneity by varying the Dirichlet parameter $\alpha$. In addition to the default $\alpha = 0.5$, we consider a near-IID setting ($\alpha = 5.0$) and a highly skewed non-IID setting ($\alpha = 0.1$). Table 5 reports mean accuracies. When the data are near-IID or mildly heterogeneous ($\alpha = 5.0$ and $0.5$), PARSE exceeds the strongest competitor by 1.1%–1.5%. Under severe heterogeneity ($\alpha = 0.1$), the margin widens to about 2.9%, indicating that feature fission and partial alignment remains effective even in the most challenging non-IID regimes.

Table 5: Comparisons across non-IID settings.

| Methods | $\alpha = 5.0$ | $\alpha = 0.5$ | $\alpha = 0.1$ |
|---|---|---|---|
| DSGD-Modality | 83.00 | 76.43 | 53.23 |
| DSGD-Task | 82.33 | 75.00 | 52.07 |
| DSGD-Hybrid | 81.33 | 75.17 | 50.67 |
| Harmony | 83.43 | 77.67 | 56.43 |
| FedHGB | 82.17 | 75.40 | 49.57 |
| DMML-KD | 81.67 | 77.00 | 55.00 |
| PARSE | 84.97 | 78.80 | 59.23 |

Table 6: Accuracy under different $\beta$ values.

| $\beta$ | Accuracy (%) | | |
|---|---|---|---|
| | Unimodal [A] | Unimodal [G] | Multimodal [AG] |
| 0.0 | 79.3±0.8 | 67.5±0.5 | 87.6±0.3 |
| 0.1 | 79.1±0.4 | 67.9±0.5 | 87.8±0.2 |
| 0.2 | 79.4±0.2 | **68.4**±0.2 | **88.6**±0.5 |
| 0.3 | **79.6**±0.5 | 67.7±0.3 | 88.4±0.6 |
| 0.4 | 78.6±0.2 | 67.2±0.7 | 87.9±0.4 |

**Effect of Contrastive Loss.** We vary the coefficient $\beta$ in Eq. 7, which scales the contrastive-diversity term for multimodal agents, from 0 to 0.4, and report accuracies in Table 6. Setting $\beta = 0.2$ consistently improves both uni- and multimodal accuracy, indicating that a moderate contrastive signal strengthens cross-modal consistency. For $\beta > 0.2$, multimodal accuracy declines because the contrastive objective begins to dominate, emphasizing modality-shared but task-irrelevant patterns and hindering supervised learning.

**Impact of Overlay Topology.** We evaluate PARSE and all baselines on three P2P overlay topologies: *Ring* (Koloskova et al., 2019); *Chordal Ring* (Ahmad et al., 2022), which augments each node with a diametrically opposite link and roughly halves the diameter; and *Random Gossip* (De Vos et al., 2023), where each round each agent exchanges with two random peers from the same group. Fig. 6 reports the average accuracy over agents. PARSE converges faster and reaches higher final accuracy than all competitors on every topology. Gains are largest on Chordal Ring and Random-Gossip, indicating that PARSE benefits from shorter path lengths and quicker consensus.

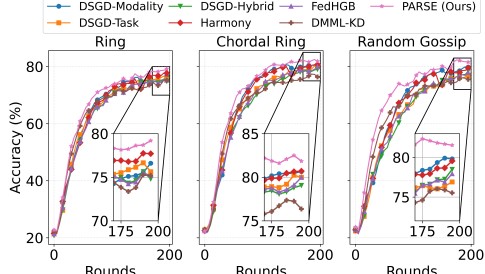

Figure 6: Comparison of different methods under various communication topologies.

**Ensemble Training Analysis.** We compare three schemes: *Ensemble-only* (ours): a single loss on the sum of all three component predictions; *Component-only*: independent losses for each component; and *Ensemble+Component*: the ensemble loss plus three per-component losses. Fig. 7 reports the average accuracy. Ensemble-only achieves the highest accuracy for both unimodal and multimodal agents. Component-only slightly reduces unimodal accuracy and substantially degrades multimodal accuracy. Adding per-component losses recovers part of the multimodal drop but hurts unimodal performance. These results indicate that an ensemble-only objective provides the most stable signal by dampening gradient noise from unaligned components.

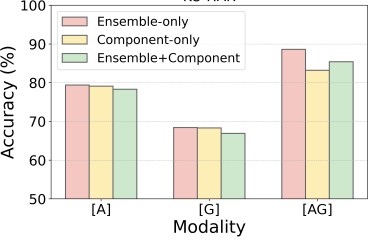

Figure 7: Impact of ensemble predictions in the classification loss.

## 5 CONCLUSION

PARSE operationalizes PID-guided feature fission with partial alignment for multimodal DFL, aligning the shareable slices while leaving irrelevant parts undisturbed. Across four benchmarks and diverse agent mixes, PARSE consistently improves both unimodal and multimodal accuracy over state-of-the-art baselines. We anticipate extensions to richer fusion and larger modality vocabularies to further broaden applicability and highlight PARSE-based promising future directions.

ETHICS STATEMENT

All datasets used in this work are publicly available research datasets, and no private or personally identifiable information is involved. Our study focuses on developing decentralized federated learning methods for multimodal data, and does not involve human subjects, interventions, or sensitive applications. We do not anticipate ethical concerns beyond those generally associated with AI research. Nonetheless, we acknowledge that biases or limitations inherent in the public datasets could influence model behavior, and encourage responsible use of our framework.

REPRODUCIBILITY STATEMENT

The experimental details of the motivation study are provided in Appendix B. Four datasets are included, and the preprocessing procedures, model architectures, and training setups are described in Appendix C. The source code and data used in our experiments will be made publicly available upon publication.

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

## A    RELATED WORK

We review three closely related research directions: multimodal representation learning, multimodal federated learning, and the emerging field of multimodal *decentralized* federated learning.

**Multimodal Representation Learning.** Early work merges modalities via simple concatenation or cross-attention, yielding a single embedding that blends redundant, unique, and synergistic cues (Gao et al., 2020; Tian et al., 2018; He et al., 2021). Later methods tighten alignment through contrastive pre-training, such as CLIP (Radford et al., 2021) and ALIGN (Jia et al., 2021), or via information-theoretic decomposition (Liang et al., 2023). But all assume centralized, full-modality access and a monolithic backbone—impractical under privacy or bandwidth constraints. In contrast, our feature-fission perspective is orthogonal and complementary: rather than learning a single embedding, we separate each latent vector into three partial information decomposition (PID)-motivated components (Liang et al., 2023). This disentanglement allows distributed agents share only the information that should be shared, while preserving modality-specific and synergistic knowledge locally.

**Multimodal Federated Learning.** Extensions of FL to multimodal data are still largely *server-centric*. Task-partitioned methods (e.g., FedPercepNet (Xiong et al., 2022), FedHGRL (Chen & Li, 2022)) cluster agents that share the same modality set and run FedAvg inside each cluster, blocking cross-cluster knowledge transfer and suffering under modality imbalance. Modality-partitioned methods (e.g., BalancedMS (Fan et al., 2024)) treat each modality as a virtual agent, so synergistic cross-modal cues remain under-utilised. Hybrid schemes (e.g., FedHGB (Chen & Li, 2022), FedCLIP (Lu et al., 2023), FedMSplit (Chen & Zhang, 2022), FedMBridge (Chen & Zhang, 2024)) blend the two ideas but still rely on a central server to resolve gradient conflicts and rebalance agents. Knowledge decomposition approaches (e.g., MCARN (Yang et al., 2024), FedHKD (Wang et al., 2024)) effectively exploit sharing among modality-heterogeneous agents; however, realizing modality interaction still requires centralized control via a dedicated server. None of the above methods tackles gradient misalignment while being simultaneously coordination-free and topology-agnostic.

**Multimodal Decentralized Federated Learning.** Eliminating the central server removes a single point of failure and alleviates privacy concerns, but it also makes cross-agent coordination substantially more difficult. DMML-KD (Yin et al., 2024) is, to the best of our knowledge, the first algorithm expressly designed for multimodal DFL. It aligns modality-shared knowledge across agents by broadcasting a feature generator over a *fully-connected* peer-to-peer network. However, it does not explicitly account for synergistic interactions and heavily relies on the presence of a sufficient number of multimodal agents (see Section 2).

## B    MOTIVATION STUDY SETUP

We evaluate each strategy on the AVE dataset (Tian et al., 2018), which contains aligned audio and video modalities for a 28-class classification task.

**Data preprocessing and allocation:** Audio signals are converted into spectrograms of size $1 \times 257 \times 1004$. For video inputs, we randomly sample 10 frames from each clip and use their mean as the representative image. All images are resized to $3 \times 224 \times 224$. We use Dirichlet distribution (Hsu et al., 2019) $Dir(\alpha = 5.0)$ to split data to a total of 30 agents.

**DNN models:** We use MobileNet-V3-Large (Howard et al., 2019) as the backbone for both audio and video modalities to extract modality-specific features in $\mathbb{R}^{256}$. For the audio input, we adapt the first convolutional layer to accept a single-channel spectrogram by modifying it to a $3 \times 3$ kernel with stride 2, padding 1, and 16 output channels (without bias). A single-layer linear classifier is used for final prediction based on the extracted features.

**Training setup:** All trainable components, including feature extractors and classifiers, are optimized using SGD with a learning rate of $5 \times 10^{-3}$, momentum of 0.9, and weight decay of $5 \times 10^{-4}$. The batch size is set to 32 across all experiments. The batch size is set to 32 across all experiments. Each agent performs one epoch of local gradient descent per communication round, for a total of 200 communication rounds.

**Fusion:** We use a simple, non-parametric averaging operation to aggregate modality-specific features into a unified fused representation. The fused features keep in $\mathbb{R}^{256}$.

**Topology and Parameter Aggregation:** We use a ring topology for all subnetworks in our experiments.

For *task-based sharing*, agents are divided into three cliques: an audio agent clique, a video agent clique, and a multimodal agent clique. Each clique forms a static ring network with randomly connected neighbors. Multimodal agents are trained using cross-entropy loss on fused features, while unimodal agents are trained on their modality-specific features. During each communication round, agents exchange all locally updated parameters with their immediate neighbors, and parameters are aggregated using uniform weights: $[1/3, 1/3, 1/3]$ for the agent and its two neighbors.

For *modality-based sharing*, agents are grouped into two modality-specific cliques (audio and video). Multimodal agents participate in both cliques and maintain separate feature extractors and classifiers for each modality. Only modality-specific parameters are shared within each clique.

For *hybrid sharing*, agents are connected as in modality-based sharing, but multimodal agents are trained with fused features, as in task-based sharing. However, only modality-specific parameters are exchanged within each clique.

# C EXPERIMENTAL SETUP

## C.1 BASIC SETUP

**Datasets.** (1) KU-HAR (Sikder & Nahid, 2021) is a human-activity-recognition benchmark with 18 actions captured by two sensors—accelerometer (A) and gyroscope (G). We perform an eight-class setting (walking, walking upstairs, walking downstairs, sitting, standing, laying, jumping, and running). (2) ModelNet-40 (Wu et al., 2015) contains CAD models from 40 object categories. We treat two rendered views of each 3-D object as distinct modalities V1, V2. (3) AVE (Tian et al., 2018) (Audio-Visual Event) comprises 10-s video clips from 28 event classes with synchronized audio and visual streams A, V. (4) IEMOCAP (Busso et al., 2008) is an emotion-recognition corpus with audio, visual, and text modalities A, V, T. Following Liang et al. (2020), we keep four emotion labels—*happy*, *sad*, *angry*, and *neutral*.

**Communication setup.** We follow the modality- and task-based connection schemes described in Section 2. Unless stated otherwise, agents are linked in a ring topology (Koloskova et al., 2019). Each ring is formed either (i) by agents that share a given modality (modality-specific ring) or (ii) by agents that have the exact same modality set (task-specific ring). During every communication round, each agent runs one local training epoch and then exchanges its modality-specific parameters only with its immediate ring neighbors.

**Baselines.** Because multimodal DFL is still nascent, we adapt several server-based multimodal FL methods to the peer-to-peer setting, selecting those that do not fundamentally depend on a central server: (1) Harmony (Ouyang et al., 2023): a two-stage scheme that we instantiate as modality-based sharing followed by task-based sharing. (2) FedHGB (Chen & Li, 2022): employs hierarchical gradient blending; we run it with a hybrid topology. (3) DMML-KD (Yin et al., 2024): uses a shared feature generator across modality-heterogeneous agents; we implement it under the same hybrid graph for fair comparison. We also include three lightweight baselines derived from DSGD (see Section 2): (4) DSGD-Modality: modality-based sharing (Yuan et al., 2024a); (5) DSGD-Task:task-based sharing (Xiong et al., 2022); (6) DSGD-Hybrid: hybrid sharing without gradient blending (Chen & Li, 2022).

## C.2 DATA PREPROCESSING

**KU-HAR.** We follow the preprocessing procedure described in Feng et al. (2023). Specifically, we preprocess the raw sensor data for both accelerometer and gyroscope signals using a standardized pipeline. The raw data, provided as CSV files, is segmented into 128-sample windows with a stride of 2, resulting in 64 time steps per segment. Each segment is then split into accelerometer (first three channels) and gyroscope (last three channels) features. For each modality, the features are normalized independently using per-segment statistics (zero mean and unit variance).

**ModelNet-40.** We load paired RGB renderings of 3D objects, where each pair consists of two views from different angles (i.e., v001 and v007 in the raw data). After loading, each image is randomly

resized and cropped to 224×224 pixels with 3 channels (RGB), and normalized using ImageNet mean and standard deviation.

**AVE.** We adopt a two-stream preprocessing strategy for both audio and visual modalities. For the **visual stream**, we randomly sample 10 frames from each clip and use their mean as the representative image. Each image is then resized and normalized using ImageNet statistics. For the **audio stream**, raw waveforms are loaded at a 16kHz sampling rate. A 10-second segment is created via tiling and trimming as needed. The waveform is then converted into a log-scaled spectrogram using the Short-Time Fourier Transform (STFT) with a window size of 512 and hop length of 353. The resulting spectrogram is normalized per segment using zero mean and unit variance.

**IEMOCAP.** We use the same preprocessing procedure as in Liang et al. (2020). We use three modalities: acoustic, visual, and textual. **Acoustic features** are extracted using the openSMILE toolkit, producing 130-dimensional frame-level descriptors. To reduce redundancy, we downsample the features by a factor of 10 and apply normalization using statistics from the training set. **Visual features** are 342-dimensional frame-level embeddings extracted using a pretrained DenseNet model from detected face regions in video frames. **Textual features** are 1024-dimensional contextual embeddings obtained from a pretrained BERT-large model, aligned at the word or token level.

### C.3    MODEL ARCHITECTURES

**KU-HAR.** As in Table 7, we use a compact temporal encoder for processing inertial signals from the KU-HAR dataset. The model consists of a 1D convolutional encoder followed by a GRU layer. The convolutional block reduces the temporal resolution while expanding the feature dimension, and the GRU captures sequential dependencies across time. A final average pooling layer aggregates temporal information into a fixed-length embedding for downstream classification.

Table 7: Architecture of the feature extractors for KU-HAR (accelerometer and gyroscope).

| Module | Output Shape | Description |
| --- | --- | --- |
| Input | $(B, 64, 3)$ | Raw inertial sequence with 64 time steps and 3-axis sensor readings. $B$ is the batch size. |
| Conv1dEncoder | $(B, 8, 128)$ | 1D convolution that reduces time from 64 to 8, expands channels to 128, with dropout = 0.1. |
| GRU | $(B, 8, 192)$ | Single-layer, unidirectional GRU with input size 128 and hidden size 192. Dropout = 0.1. |
| Average Pooling | $(B, 192)$ | Temporal average pooling across 8 time steps for fixed-length output. |

**ModelNet-40.** We adopt a lightweight convolutional backbone based on MobileNetV3-Small (Howard et al., 2019) for visual feature extraction. Specifically, we use a pre-trained MobileNetV3-Small model and replace its original classifier with a single linear layer that projects the output to a 300-dimensional embedding space.

**AVE.** To accommodate single-channel audio spectrogram inputs, we modify the first convolutional layer of MobileNetV3-Large. The original model is designed for 3-channel RGB inputs and begins with a convolutional layer defined as Conv2d(input_channels=3, output_channels=16, kernel_size=3, stride=2, padding=1). We replace this with Conv2d(input_channels=1, output_channels=16, kernel_size=3, stride=2, padding=1), keeping all other parameters unchanged. This adjustment enables the model to process grayscale spectrograms directly. The output will be in a 384-dimensional embedding space.

For the visual stream in AVE, we use a MobileNetV3-Large backbone pre-trained on ImageNet to extract frame-level visual embeddings. The original classification head is replaced with a single linear layer that projects the features to a 384-dimensional embedding space.

**IEMOCAP.** As in Table 8, we design modality-specific encoders tailored to the characteristics of audio, video, and text inputs. For audio, frame-level acoustic features (130-dimensional) are processed using a single-layer unidirectional LSTM with a hidden size of 384, followed by temporal max pooling to obtain a fixed-length audio embedding. For video, visual features (342-dimensional per frame) are encoded using the same LSTM-based architecture as audio, capturing temporal facial dynamics. Max pooling is applied over the sequence to produce the video embedding. Utterance-level contextual embeddings from a pre-trained BERT model (1024-dimensional) are passed through three parallel 1D convolutional branches with kernel sizes of 3, 4, and 5. The outputs are concatenated,

regularized with dropout (0.5), and projected through a fully connected layer with ReLU activation to obtain a 384-dimensional text embedding.

Table 8: Architectures of the feature extractors for IEMOCAP (where $B$ is the batch size).

| Modality | Module | Output Shape | Description |
|---|---|---|---|
| Audio | Input | $(B, T, 130)$ | Frame-level acoustic features with 130 dimensions. |
| | LSTM | $(B, T, 384)$ | Single-layer, unidirectional LSTM with hidden size 384. |
| | Max Pooling | $(B, 384)$ | Temporal max pooling over $T$ time steps. |
| Video | Input | $(B, T, 342)$ | Frame-level features (342 dimensions). |
| | LSTM | $(B, T, 384)$ | Single-layer, unidirectional LSTM with hidden size 384. |
| | Max Pooling | $(B, 384)$ | Temporal max pooling over $T$ time steps. |
| Text | Input | $(B, T, 1024)$ | BERT embeddings (1024 dimensions) over $T$ segments. |
| | Conv2D (3 branches) | $(B, 128) \times 3$ | 1D convolutions with kernel sizes 3, 4, and 5. |
| | Concat + Dropout | $(B, 384)$ | Concatenation followed by dropout with $p = 0.5$. |
| | FC + ReLU | $(B, 384)$ | Final projection with ReLU activation. |

## C.4 TRAINING SETUP

**Datasets.** For **KU-HAR**, **ModelNet40**, and **AVE**, we use stochastic gradient descent (SGD) with momentum of 0.9 and weight decay of $5 \times 10^{-4}$. The learning rates are set to 0.1, 0.01, and 0.005, respectively. The batch sizes are 64 for KU-HAR and ModelNet40, and 32 for AVE. For **IEMOCAP**, we adopt the Adam optimizer with a learning rate of $5 \times 10^{-5}$ and $\beta$ values of (0.9, 0.999), using a batch size of 64. By default, we split each dataset into 80% for training and 20% for testing.

**Hardware.** All experiments are conducted on a machine equipped with a Tesla V100-SXM2 GPU (32GB VRAM) and a Intel Xeon Silver 4216 CPU (2.1 GHz, 16 cores, 22MB cache).

**Methods.** For PARSE, we decompose the input features into three components. The feature dimensions are split from 384 to 128 on IEMOCAP and AVE, from 300 to 100 on ModelNet40, and from 192 to 64 on KU-HAR. The temperature hyperparameter for the contrastive loss is set to 0.2 throughout all experiments. For **Harmony**, we follow a two-stage training strategy: 150 rounds of modality-independent training followed by 50 rounds of modality-joint training. For **FedHGB**, we reserve 10% of the training data as a validation set. For **DMML-KD**, we adopt the same feature aggregator as used in Zhu et al. (2021).

# D ADDITIONAL RESULTS

## D.1 ADDITIONAL RESULTS FOR VARYING NON-IID CONFIGURATIONS

To provide a more fine-grained analysis of our method under varying non-IID conditions, we conduct additional experiments comparing agent ratios across different levels of heterogeneity. These results, presented in Table 9 and Table 10, complement the main findings in Section 4. Across all evaluated configurations, PARSE consistently achieves the highest accuracy, demonstrating robust performance for multimodal agents and unimodal agents across varying agent ratios.

When the non-IID level is low ($\alpha = 5.0$), PARSE outperforms baseline methods by a notable margin. For instance, it surpasses other methods by at least 3.0% on modality V2 of the ModelNet-40 dataset, when the proportion of unimodal agents is small. On the IEMOCAP dataset, PARSE improves multimodal performance by approximately 1.2–2.5% across various settings, demonstrating its effectiveness in learning modality-interactive features. Even under settings where baseline methods struggle, PARSE continues to deliver substantial performance gains, highlighting its robustness to agent heterogeneity and adverse modality configurations. At a higher non-IID level ($\alpha = 0.1$), the improvements become even more pronounced. PARSE yields significant accuracy boosts for multimodal agents, particularly on the ModelNet-40 and AVE datasets—further suggesting its strength in highly heterogeneous environments.

The results confirm that our method benefits both multimodal and unimodal agents simultaneously. The proposed **feature fission** and **partial alignment** mechanisms facilitate effective knowledge shar-

Table 9: Performance (accuracy %) of methods under varying agent-ratio scenarios (Dirichlet $\alpha = 5.0$, ring topology).

| Agent ratios | | 6 : 6 : 18 | | | 10 : 10 : 10 | | | 13 : 13 : 4 | | |
|---|---|---|---|---|---|---|---|---|---|---|
| **Agent types** | | [A] | [G] | [AG] | [A] | [G] | [AG] | [A] | [G] | [AG] |
| KU-HAR | DSGD-Modality | 84.5±1.0 | 73.2±0.9 | 88.8±0.7 | 84.3±0.1 | 75.0±0.5 | 89.7±0.2 | 83.9±0.6 | 77.4±0.7 | 90.5±0.4 |
| | DSGD-Task | 82.8±0.9 | 71.5±1.9 | 88.2±0.7 | 84.0±0.3 | 74.7±0.4 | 88.3±0.2 | 84.0±0.6 | 76.4±0.8 | 86.8±0.5 |
| | DSGD-Hybrid | 82.5±1.2 | 69.0±1.0 | 88.5±0.7 | 82.6±0.8 | 72.1±0.6 | 89.3±0.9 | 82.9±0.5 | 74.1±0.8 | 90.8±0.4 |
| | Harmony | 84.1±0.7 | 72.9±0.7 | 91.0±0.4 | 84.5±0.3 | 75.0±0.4 | 90.8±0.2 | 84.6±1.0 | 77.5±0.9 | 90.8±0.4 |
| | FedHGB | 82.5±0.8 | 71.7±1.2 | 88.1±0.6 | 84.0±0.8 | 73.7±1.0 | 88.8±1.2 | 81.4±0.6 | 74.3±1.5 | 90.1±0.3 |
| | DMML-KD | 84.5±0.4 | 68.7±1.0 | 89.8±0.7 | 84.4±1.0 | 69.6±1.1 | 91.0±0.9 | 82.6±1.2 | 77.0±0.4 | 91.0±0.5 |
| | PARSE | 84.5±1.4 | 78.6±1.0 | 91.2±0.8 | 86.4±0.9 | 77.1±0.4 | 91.4±0.3 | 84.6±1.2 | 78.3±1.3 | 91.5±0.8 |
| **Agent types** | | [V1] | [V2] | [V1,V2] | [V1] | [V2] | [V1,V2] | [V1] | [V2] | [V1,V2] |
| ModelNet-40 | DSGD-Modality | 77.4±0.4 | 73.2±1.2 | 81.4±0.5 | 77.2±1.4 | 69.6±0.8 | 81.4±1.4 | 77.2±1.2 | 75.2±0.8 | 76.3±0.9 |
| | DSGD-Task | 77.0±0.9 | 66.2±1.4 | 77.5±1.3 | 76.4±1.5 | 66.4±0.7 | 76.3±1.3 | 75.4±1.2 | 76.3±0.8 | 75.3±0.6 |
| | DSGD-Hybrid | 76.4±0.4 | 70.3±1.2 | 74.5±0.3 | 76.9±1.8 | 67.5±1.4 | 81.3±1.3 | 77.0±0.6 | 68.9±1.3 | 78.6±0.6 |
| | Harmony | 76.5±1.2 | 72.1±1.2 | 81.6±0.7 | 77.1±1.3 | 75.2±1.8 | 81.0±1.2 | 77.2±0.7 | 76.4±0.3 | 80.6±0.5 |
| | FedHGB | 76.9±0.5 | 71.0±1.3 | 81.6±1.8 | 73.4±1.3 | 70.4±2.3 | 81.4±1.5 | 77.5±0.7 | 72.7±1.5 | 80.1±0.9 |
| | DMML-KD | 77.4±0.8 | 71.8±2.0 | 80.8±1.2 | 76.6±1.0 | 68.8±1.9 | 80.8±1.0 | 76.2±0.8 | 71.1±1.4 | 78.2±0.7 |
| | PARSE | 81.1±0.6 | 76.3±1.0 | 82.7±0.4 | 77.5±1.0 | 76.3±1.2 | 82.4±1.1 | 78.3±1.6 | 76.5±0.7 | 80.8±0.7 |
| **Agent types** | | [A] | [V] | [AV] | [A] | [V] | [AV] | [A] | [V] | [AV] |
| AVE | DSGD-Modality | 50.2±0.9 | 56.6±0.5 | 68.1±1.2 | 51.6±0.6 | 56.0±0.6 | 67.8±0.2 | 50.2±0.3 | 55.3±0.4 | 69.0±0.5 |
| | DSGD-Task | 46.2±1.4 | 51.4±0.4 | 65.3±1.0 | 49.3±0.5 | 51.9±0.4 | 61.5±0.2 | 49.3±0.4 | 55.3±0.6 | 57.0±0.7 |
| | DSGD-Hybrid | 46.3±0.5 | 52.7±0.9 | 66.4±0.6 | 50.3±0.8 | 54.1±0.3 | 68.5±0.2 | 50.4±1.4 | 54.8±0.5 | 69.4±1.0 |
| | Harmony | 49.2±0.6 | 56.2±0.9 | 69.2±1.9 | 51.0±0.1 | 54.7±0.2 | 70.8±1.0 | 51.1±0.7 | 55.2±0.6 | 69.4±1.1 |
| | FedHGB | 49.2±1.5 | 55.4±0.6 | 69.2±1.0 | 50.6±0.8 | 56.0±0.8 | 66.7±0.7 | 51.7±0.4 | 55.7±0.8 | 69.6±0.6 |
| | DMML-KD | 47.8±0.4 | 56.7±0.5 | 69.4±1.1 | 49.5±0.6 | 52.2±1.3 | 68.3±1.5 | 51.0±0.8 | 54.2±0.4 | 69.6±0.7 |
| | PARSE | 50.4±0.7 | 57.0±0.4 | 69.4±0.7 | 51.9±1.7 | 56.7±1.3 | 71.4±1.6 | 52.1±1.1 | 55.8±1.7 | 69.8±0.6 |

| Agent ratios | | 6 : 6 : 6 : 22 | | | | 10 : 10 : 10 : 10 | | | | 12 : 12 : 12 : 4 | | | |
|---|---|---|---|---|---|---|---|---|---|---|---|---|---|
| **Agent types** | | [A] | [V] | [T] | [AVT] | [A] | [V] | [T] | [AVT] | [A] | [V] | [T] | [AVT] |
| IEMOCAP | DSGD-Modality | 51.2±0.2 | 56.8±1.1 | 61.2±0.4 | 74.3±0.2 | 51.1±0.5 | 56.8±0.4 | 62.5±0.4 | 74.0±0.7 | 51.0±0.5 | 56.2±0.7 | 62.2±0.3 | 72.1±0.4 |
| | DSGD-Task | 50.2±0.5 | 53.6±0.8 | 56.3±0.2 | 72.6±0.2 | 50.7±0.4 | 57.1±0.4 | 60.3±0.5 | 69.8±0.2 | 52.4±0.2 | 56.6±0.3 | 61.7±0.3 | 64.1±0.5 |
| | DSGD-Hybrid | 50.7±1.0 | 49.9±0.4 | 61.5±0.4 | 73.6±0.3 | 50.2±0.7 | 56.7±0.5 | 62.1±0.9 | 73.5±0.7 | 52.3±0.6 | 56.7±0.5 | 62.4±0.2 | 72.6±0.4 |
| | Harmony | 50.4±0.7 | 54.3±0.6 | 60.1±0.4 | 74.2±0.2 | 50.9±0.8 | 56.8±0.3 | 62.4±0.1 | 73.6±0.5 | 52.4±0.6 | 56.3±0.4 | 62.4±0.5 | 69.9±0.3 |
| | FedHGB | 51.0±0.4 | 55.2±0.8 | 60.1±0.7 | 75.1±0.3 | 50.4±1.0 | 56.1±0.7 | 61.8±0.5 | 74.3±1.2 | 52.1±0.3 | 56.2±0.5 | 62.4±0.3 | 74.4±0.6 |
| | DMML-KD | 50.9±0.2 | 49.3±0.4 | 60.8±0.8 | 75.3±0.3 | 51.9±1.4 | 54.3±1.1 | 61.7±0.3 | 74.8±0.8 | 53.2±0.5 | 55.1±0.8 | 62.5±0.3 | 74.5±0.6 |
| | PARSE | 53.0±0.4 | 57.0±1.0 | 62.0±0.5 | 76.5±0.3 | 53.2±0.7 | 57.3±0.4 | 63.3±0.3 | 77.2±0.5 | 53.9±0.3 | 56.9±0.5 | 62.7±0.2 | 76.7±0.4 |

Table 10: Performance (accuracy %) of methods under varying agent-ratio scenarios (Dirichlet $\alpha = 0.1$, ring topology).

| Agent ratios | | 6 : 6 : 18 | | | 10 : 10 : 10 | | | 13 : 13 : 4 | | |
|---|---|---|---|---|---|---|---|---|---|---|
| **Agent types** | | [A] | [G] | [AG] | [A] | [G] | [AG] | [A] | [G] | [AG] |
| KU-HAR | DSGD-Modality | 57.4±1.4 | 48.3±0.8 | 52.3±1.3 | 57.9±0.3 | 45.8±0.4 | 56.0±1.1 | 58.3±1.5 | 44.2±0.9 | 61.4±1.2 |
| | DSGD-Task | 39.7±1.8 | 47.1±2.1 | 60.4±0.9 | 49.3±0.2 | 45.5±0.5 | 61.4±0.3 | 55.3±1.2 | 44.6±1.7 | 64.5±1.4 |
| | DSGD-Hybrid | 42.9±0.8 | 42.5±1.2 | 56.7±1.1 | 48.6±0.9 | 41.2±0.8 | 62.2±1.3 | 51.3±1.8 | 41.5±0.6 | 62.4±2.0 |
| | Harmony | 56.3±2.1 | 47.7±1.5 | 59.7±1.5 | 55.8±0.3 | 44.7±0.4 | 68.8±0.3 | 57.6±1.6 | 44.2±1.7 | 66.7±1.2 |
| | FedHGB | 43.9±0.8 | 42.6±1.1 | 48.6±1.3 | 46.2±0.7 | 42.4±1.0 | 60.1±0.8 | 56.2±1.0 | 43.1±1.6 | 64.8±1.2 |
| | DMML-KD | 54.0±1.2 | 53.8±1.5 | 60.5±0.9 | 56.7±0.6 | 40.3±0.5 | 68.0±0.7 | 58.8±1.7 | 44.3±2.5 | 67.7±2.3 |
| | PARSE | 58.7±0.9 | 54.0±1.3 | 60.9±1.7 | 60.4±0.7 | 46.6±0.4 | 70.7±0.9 | 59.2±1.4 | 46.8±1.3 | 68.1±0.8 |
| **Agent types** | | [V1] | [V2] | [V1,V2] | [V1] | [V2] | [V1,V2] | [V1] | [V2] | [V1,V2] |
| ModelNet-40 | DSGD-Modality | 52.3±1.6 | 37.4±1.4 | 51.2±1.1 | 46.2±0.8 | 44.8±1.2 | 55.7±1.3 | 41.6±1.6 | 42.4±1.2 | 54.2±1.5 |
| | DSGD-Task | 36.8±1.6 | 35.1±1.3 | 43.5±2.4 | 42.7±1.4 | 39.8±1.5 | 49.3±0.9 | 39.5±2.6 | 41.9±1.6 | 39.4±1.1 |
| | DSGD-Hybrid | 36.7±1.8 | 32.9±2.3 | 44.7±1.6 | 36.9±0.7 | 34.8±1.2 | 48.4±0.5 | 39.3±1.1 | 42.9±1.5 | 48.1±1.4 |
| | Harmony | 49.9±1.4 | 34.2±0.9 | 51.8±2.0 | 44.2±0.9 | 43.7±0.7 | 54.5±1.3 | 40.4±2.7 | 42.2±1.7 | 44.8±2.2 |
| | FedHGB | 41.4±1.7 | 33.2±1.6 | 47.2±2.2 | 40.6±1.6 | 34.0±1.4 | 47.8±2.0 | 41.1±2.6 | 39.3±1.8 | 44.1±2.6 |
| | DMML-KD | 43.6±2.0 | 35.4±2.3 | 56.2±2.1 | 45.1±1.6 | 41.3±2.7 | 54.4±2.3 | 44.5±1.2 | 34.8±1.8 | 61.0±1.8 |
| | PARSE | 55.3±1.9 | 42.3±1.6 | 60.7±1.7 | 53.6±1.3 | 45.0±0.4 | 62.5±0.5 | 46.5±1.7 | 47.1±1.9 | 65.4±1.6 |
| **Agent types** | | [A] | [V] | [AV] | [A] | [V] | [AV] | [A] | [V] | [AV] |
| AVE | DSGD-Modality | 30.4±1.4 | 39.3±1.5 | 43.3±1.5 | 34.4±1.3 | 38.4±1.0 | 44.5±0.7 | 32.6±1.5 | 36.4±0.9 | 46.7±0.8 |
| | DSGD-Task | 27.4±1.0 | 33.6±1.4 | 38.0±0.9 | 32.1±1.0 | 37.5±1.1 | 35.3±1.1 | 30.4±1.4 | 35.5±2.0 | 25.6±0.9 |
| | DSGD-Hybrid | 25.7±1.1 | 34.9±1.3 | 42.5±0.7 | 31.3±1.6 | 35.8±0.7 | 43.0±0.6 | 29.5±1.2 | 34.9±1.5 | 36.8±2.1 |
| | Harmony | 29.9±1.2 | 37.8±0.9 | 41.9±1.5 | 32.2±1.5 | 39.3±0.6 | 42.4±0.8 | 31.7±0.8 | 36.4±1.0 | 48.0±1.1 |
| | FedHGB | 27.3±1.6 | 37.4±1.9 | 44.8±1.5 | 31.6±2.0 | 38.7±0.8 | 44.7±1.3 | 30.7±1.2 | 37.1±1.4 | 39.0±0.7 |
| | DMML-KD | 28.7±0.9 | 37.4±1.0 | 46.3±1.8 | 31.9±0.5 | 31.8±0.6 | 47.5±1.3 | 29.7±1.9 | 37.2±1.7 | 51.3±1.1 |
| | PARSE | 31.1±0.8 | 40.7±1.0 | 46.6±1.5 | 34.9±1.4 | 39.5±0.9 | 51.8±0.8 | 33.6±0.8 | 39.2±1.3 | 52.7±1.4 |

| Agent ratios | | 6 : 6 : 6 : 22 | | | | 10 : 10 : 10 : 10 | | | | 12 : 12 : 12 : 4 | | | |
|---|---|---|---|---|---|---|---|---|---|---|---|---|---|
| **Agent types** | | [A] | [V] | [T] | [AVT] | [A] | [V] | [T] | [AVT] | [A] | [V] | [T] | [AVT] |
| IEMOCAP | DSGD-Modality | 35.5±1.3 | 41.2±1.1 | 54.1±1.6 | 49.8±1.6 | 37.4±0.9 | 50.1±0.7 | 48.0±0.9 | 51.5±1.7 | 37.5±1.7 | 45.1±1.2 | 47.6±0.8 | 63.0±0.5 |
| | DSGD-Task | 34.5±1.2 | 37.8±1.2 | 46.4±0.5 | 55.4±1.3 | 38.1±0.5 | 49.1±0.4 | 47.6±0.5 | 54.4±0.3 | 37.0±2.3 | 46.5±1.3 | 48.4±0.7 | 52.7±0.4 |
| | DSGD-Hybrid | 34.7±1.4 | 40.4±1.2 | 55.8±1.3 | 55.9±0.6 | 35.4±1.5 | 49.4±0.9 | 50.9±0.8 | 56.8±0.5 | 37.1±1.9 | 45.6±0.7 | 48.0±0.9 | 61.2±0.8 |
| | Harmony | 35.3±2.1 | 40.8±0.9 | 54.8±2.3 | 52.7±1.7 | 36.1±0.7 | 49.5±1.8 | 48.7±1.5 | 53.2±0.9 | 37.6±1.5 | 46.5±2.3 | 48.5±1.4 | 62.9±0.8 |
| | FedHGB | 36.2±2.0 | 41.3±1.9 | 55.3±1.6 | 49.3±1.1 | 36.4±1.7 | 49.1±1.2 | 46.7±1.3 | 53.0±0.8 | 38.4±1.2 | 45.6±0.8 | 47.0±0.9 | 61.2±0.6 |
| | DMML-KD | 36.0±1.3 | 40.6±1.0 | 56.0±1.9 | 60.3±2.1 | 36.1±1.0 | 48.0±0.4 | 48.6±0.6 | 61.5±1.4 | 35.6±1.3 | 45.0±0.9 | 47.9±1.3 | 69.8±1.0 |
| | PARSE | 36.7±1.8 | 41.8±1.5 | 56.0±0.7 | 60.5±0.6 | 38.4±0.4 | 50.6±1.1 | 51.7±1.8 | 65.1±0.7 | 38.7±1.7 | 47.1±0.6 | 48.8±0.5 | 70.4±1.2 |

ing across modality-heterogeneous agents and mitigate gradient misalignment, while the **contrastive loss** further distills cross-modality shared features to enhance the performance of unimodal agents.

Table 11: Performance (accuracy %) of methods under varying agent-ratio scenarios (Dirichlet $\alpha = 0.5$, Chordal Ring topology).

| Agent ratios | | 6 : 6 : 18 | | | 10 : 10 : 10 | | | 13 : 13 : 4 | | |
|---|---|---|---|---|---|---|---|---|---|---|
| **Agent types** | | **[A]** | **[G]** | **[AG]** | **[A]** | **[G]** | **[AG]** | **[A]** | **[G]** | **[AG]** |
| KU-HAR | DSGD-Modality | 82.2±1.0 | 75.6±1.5 | 85.4±1.2 | 82.0±0.5 | 75.8±0.9 | 88.2±0.8 | 81.2±1.3 | 76.1±1.3 | 88.0±1.2 |
| | DSGD-Task | 77.8±0.9 | 67.5±1.2 | 84.5±0.9 | 82.0±1.8 | 72.4±0.9 | 84.0±1.5 | 81.5±0.7 | 74.7±0.6 | 80.2±2.0 |
| | DSGD-Hybrid | 77.1±0.6 | 62.4±0.8 | 85.0±1.6 | 78.2±1.5 | 70.2±0.7 | 87.9±0.5 | 80.2±0.9 | 75.5±0.8 | 88.7±0.7 |
| | Harmony | 81.9±1.0 | 75.2±1.9 | 91.0±0.4 | 81.0±1.7 | 74.5±0.4 | 90.4±0.9 | 80.6±1.5 | 75.9±0.8 | 89.7±1.0 |
| | FedHGB | 77.5±0.7 | 70.2±0.8 | 85.1±0.6 | 79.3±1.0 | 72.8±0.5 | 88.0±0.9 | 80.7±0.5 | 77.1±0.4 | 88.3±1.0 |
| | DMML-KD | 80.4±1.5 | 67.1±2.0 | 88.2±0.6 | 81.5±1.1 | 70.2±1.7 | 89.3±0.7 | 82.1±1.4 | 75.4±1.5 | 90.0±1.7 |
| | PARSE | 82.4±1.2 | 76.1±1.3 | 91.8±0.5 | 82.6±1.1 | 76.5±1.8 | 90.8±0.9 | 83.4±0.7 | 78.4±0.8 | 90.3±1.2 |
| **Agent types** | | **[V1]** | **[V2]** | **[V1,V2]** | **[V1]** | **[V2]** | **[V1,V2]** | **[V1]** | **[V2]** | **[V1,V2]** |
| ModelNet-40 | DSGD-Modality | 80.9±1.0 | 72.8±0.8 | 81.6±1.1 | 78.1±0.5 | 71.9±1.3 | 80.8±1.0 | 80.3±1.0 | 72.9±2.1 | 82.6±1.3 |
| | DSGD-Task | 71.1±0.7 | 69.1±1.1 | 75.4±0.7 | 75.6±2.1 | 71.2±0.7 | 71.7±0.6 | 76.4±0.9 | 71.6±1.3 | 66.1±1.6 |
| | DSGD-Hybrid | 72.2±1.5 | 63.3±1.6 | 74.7±1.3 | 74.8±0.8 | 67.3±1.1 | 77.3±0.7 | 78.8±2.1 | 71.2±1.7 | 80.1±1.2 |
| | Harmony | 78.8±1.3 | 71.7±1.4 | 81.2±0.4 | 80.1±1.1 | 75.2±1.3 | 81.0±0.8 | 77.1±0.6 | 71.5±1.9 | 79.0±1.7 |
| | FedHGB | 72.0±0.7 | 65.5±1.2 | 77.2±2.1 | 78.6±0.9 | 71.6±1.1 | 77.6±0.5 | 77.3±0.8 | 73.1±1.8 | 79.3±1.1 |
| | DMML-KD | 79.6±1.2 | 59.3±1.1 | 79.7±0.9 | 80.4±1.1 | 57.8±1.5 | 81.3±1.1 | 78.9±1.5 | 69.1±1.7 | 82.2±0.9 |
| | PARSE | 81.6±1.2 | 76.4±0.9 | 82.1±0.6 | 80.9±0.5 | 76.0±0.8 | 81.6±0.7 | 81.4±1.3 | 75.2±1.5 | 83.0±0.7 |
| **Agent types** | | **[A]** | **[V]** | **[AV]** | **[A]** | **[V]** | **[AV]** | **[A]** | **[V]** | **[AV]** |
| AVE | DSGD-Modality | 47.2±1.2 | 53.8±1.7 | 67.5±1.1 | 46.6±1.4 | 53.3±1.4 | 65.3±1.0 | 45.1±1.6 | 52.2±1.2 | 64.9±1.5 |
| | DSGD-Task | 38.5±0.9 | 47.0±0.8 | 61.5±0.7 | 40.7±1.0 | 50.8±0.5 | 57.7±1.5 | 42.7±1.8 | 51.5±1.8 | 51.3±0.6 |
| | DSGD-Hybrid | 39.1±0.6 | 50.2±0.9 | 64.7±0.8 | 42.6±0.5 | 51.6±0.8 | 65.4±0.7 | 44.3±1.6 | 52.6±1.3 | 65.6±1.6 |
| | Harmony | 42.3±0.8 | 51.2±1.4 | 65.6±1.2 | 45.0±0.7 | 51.3±1.6 | 66.6±1.3 | 45.2±0.5 | 51.0±1.5 | 64.3±1.0 |
| | FedHGB | 46.2±0.6 | 53.6±0.6 | 67.1±0.5 | 46.5±0.8 | 53.7±1.1 | 66.1±0.7 | 46.8±0.9 | 53.4±1.1 | 66.4±1.3 |
| | DMML-KD | 43.3±1.7 | 52.7±1.2 | 67.0±0.6 | 44.3±1.5 | 53.2±1.3 | 66.7±0.6 | 45.8±1.2 | 52.8±1.1 | 67.3±0.6 |
| | PARSE | 47.6±1.1 | 55.4±1.3 | 68.1±0.5 | 47.0±0.5 | 54.3±0.7 | 67.2±0.8 | 47.1±0.3 | 53.7±1.0 | 67.3±0.6 |

| Agent ratios | | 6 : 6 : 6 : 22 | | | | 10 : 10 : 10 : 10 | | | | 12 : 12 : 12 : 4 | | | |
|---|---|---|---|---|---|---|---|---|---|---|---|---|---|
| **Agent types** | | **[A]** | **[V]** | **[T]** | **[AVT]** | **[A]** | **[V]** | **[T]** | **[AVT]** | **[A]** | **[V]** | **[T]** | **[AVT]** |
| IEMOCAP | DSGD-Modality | 47.5±0.6 | 48.8±1.6 | 62.6±0.8 | 70.4±0.2 | 46.3±0.5 | 52.2±0.8 | 60.2±0.7 | 70.3±0.2 | 51.1±0.3 | 53.1±0.4 | 58.0±0.4 | 71.8±0.6 |
| | DSGD-Task | 48.5±0.8 | 45.1±0.6 | 58.5±0.4 | 72.3±0.6 | 44.8±0.5 | 51.9±2.1 | 58.7±0.7 | 70.1±0.5 | 51.1±0.3 | 52.7±0.4 | 58.5±0.8 | 64.4±1.1 |
| | DSGD-Hybrid | 47.3±0.6 | 45.6±0.4 | 62.3±0.6 | 71.9±0.8 | 48.0±1.3 | 51.0±0.5 | 60.6±0.8 | 71.2±0.6 | 51.0±0.4 | 53.0±1.3 | 58.1±0.6 | 71.2±0.4 |
| | Harmony | 47.4±1.2 | 47.3±0.8 | 63.3±0.2 | 72.4±1.0 | 48.3±1.5 | 51.9±0.6 | 60.1±0.5 | 72.4±0.3 | 50.9±0.6 | 53.0±1.8 | 58.3±1.8 | 70.9±0.4 |
| | FedHGB | 47.4±0.9 | 48.5±1.6 | 62.5±0.3 | 70.3±0.5 | 46.4±0.7 | 50.2±1.0 | 60.2±0.5 | 70.7±0.5 | 51.4±0.4 | 52.0±1.7 | 57.8±0.6 | 71.2±0.8 |
| | DMML-KD | 48.0±0.3 | 45.1±0.6 | 63.1±0.3 | 74.3±0.2 | 47.7±1.3 | 49.2±0.5 | 58.3±0.7 | 74.7±0.5 | 51.4±1.5 | 50.5±0.3 | 57.9±0.6 | 74.1±0.6 |
| | PARSE | 48.0±0.3 | 52.0±0.4 | 64.3±0.5 | 75.8±0.5 | 48.5±0.2 | 52.6±0.4 | 60.7±0.8 | 74.7±0.2 | 51.7±0.5 | 53.2±0.5 | 58.7±0.9 | 74.3±0.4 |

## D.2 Additional Results for Varying Topology Settings

We also conduct experiments under the default non-IID setting ($\alpha = 0.5$) using different topology configurations and agent ratios. All methods are trained for 200 rounds, and the final test accuracies are reported in Table 11 and Table 12.

The observations under the Chordal Ring topology (Table 11) are consistent with those in Section 4. PARSE outperforms both DSGD-Modality and Harmony—two methods that specialize in unimodal and multimodal agents, respectively. While DSGD-Modality often achieves strong performance on unimodal agents, and Harmony excels on multimodal ones, PARSE achieves state-of-the-art accuracy across all agent types and ratios.

The results demonstrate that PARSE does not rely on a specific communication topology between agents; instead, it maintains strong performance across different topological configurations.

## D.3 Modality-Mix Results on IEMOCAP

As a complement to the main results in Section 4, we further evaluate performance across a broader set of modality combinations on the IEMOCAP dataset. Specifically, we consider all possible subsets of the three modalities: [A], [V], [T], [AV], [AT], [VT], and [AVT], covering a total of seven distinct modality sets. To simulate varying degrees of modality availability, we experiment with different agent ratio configurations, which correspond to the following settings:

- **(30%:30%:40%)**. *We select 30% of agents to miss two modalities (i.e., unimodal agents), 30% to miss one modality (i.e., bimodal agents), and 40% to have full modality access.*

- **(43%:43%:14%)**. *We select 43% of agents to miss two modalities, 43% to miss one modality, and 14% to have full modality access.*

Table 12: Performance (accuracy %) of methods under varying agent-ratio scenarios (Dirichlet $\alpha = 0.5$, Random Gossip Topology).

| Agent ratios | 6 : 6 : 18 | | | 10 : 10 : 10 | | | 13 : 13 : 4 | | |
|---|---|---|---|---|---|---|---|---|---|
| **Agent types** | **[A]** | **[G]** | **[AG]** | **[A]** | **[G]** | **[AG]** | **[A]** | **[G]** | **[AG]** |
| DSGD-Modality | 81.2±1.7 | 74.3±1.1 | 86.1±0.6 | 80.5±1.5 | 72.6±1.8 | 85.8±1.3 | 79.2±1.1 | 72.4±2.1 | 84.9±1.2 |
| DSGD-Task | 74.6±0.8 | 63.9±1.7 | 83.9±0.6 | 77.8±1.9 | 68.4±1.6 | 83.4±0.8 | 77.2±1.0 | 69.4±2.5 | 77.4±0.7 |
| DSGD-Hybrid | 76.2±0.8 | 64.5±1.0 | 85.7±0.4 | 78.5±1.5 | 70.8±1.3 | 86.3±1.4 | 78.9±0.5 | 71.9±0.7 | 87.1±1.7 |
| Harmony | 81.0±0.9 | 73.2±1.1 | 87.5±1.7 | 80.4±1.3 | 72.5±1.4 | 86.4±1.8 | 79.0±1.5 | 72.8±2.0 | 85.2±1.2 |
| FedHGB | 78.3±1.7 | 69.2±1.0 | 85.2±1.3 | 79.6±1.5 | 71.8±1.0 | 86.1±0.7 | 77.7±1.2 | 73.4±1.3 | 86.7±0.8 |
| DMML-KD | 80.6±1.6 | 62.6±1.5 | 87.3±1.3 | 80.2±1.9 | 68.2±1.6 | 87.3±1.7 | 80.4±1.7 | 73.2±0.8 | 86.1±2.2 |
| PARSE | 82.2±0.8 | 77.2±1.6 | 88.0±1.1 | 80.5±1.9 | 76.4±1.5 | 89.7±0.7 | 80.7±0.6 | 75.7±1.7 | 87.4±1.2 |
| **Agent types** | **[V1]** | **[V2]** | **[V1,V2]** | **[V1]** | **[V2]** | **[V1,V2]** | **[V1]** | **[V2]** | **[V1,V2]** |
| DSGD-Modality | 82.4±1.7 | 78.4±1.2 | 83.0±1.4 | 80.6±1.6 | 76.4±0.5 | 84.1±0.6 | 80.4±1.2 | 77.7±0.8 | 82.1±0.7 |
| DSGD-Task | 74.4±1.8 | 69.3±0.9 | 78.1±1.8 | 78.8±1.6 | 72.0±1.5 | 77.6±1.5 | 79.2±0.9 | 74.9±1.9 | 67.5±1.3 |
| DSGD-Hybrid | 79.0±0.9 | 70.3±2.5 | 79.7±1.6 | 79.7±1.9 | 72.7±1.7 | 78.3±1.3 | 80.2±0.8 | 74.6±1.0 | 79.4±1.2 |
| Harmony | 81.2±0.6 | 77.1±1.4 | 83.8±0.5 | 79.0±1.7 | 74.9±1.2 | 83.7±0.7 | 80.4±1.0 | 77.3±1.1 | 80.6±1.0 |
| FedHGB | 80.1±0.9 | 72.0±1.3 | 80.5±0.6 | 79.7±1.6 | 72.4±0.6 | 78.8±0.9 | 78.7±1.7 | 73.5±2.2 | 78.6±1.3 |
| DMML-KD | 80.8±1.7 | 67.7±1.9 | 81.2±1.7 | 80.7±1.3 | 68.9±1.4 | 81.8±1.6 | 81.2±0.8 | 71.5±1.3 | 83.4±1.0 |
| PARSE | 84.0±0.8 | 81.4±0.7 | 84.4±0.5 | 82.8±1.4 | 78.2±1.1 | 84.9±1.8 | 82.4±0.8 | 78.3±1.4 | 84.2±0.7 |
| **Agent types** | **[A]** | **[V]** | **[AV]** | **[A]** | **[V]** | **[AV]** | **[A]** | **[V]** | **[AV]** |
| DSGD-Modality | 51.6±1.2 | 55.8±0.4 | 67.8±1.1 | 51.2±0.6 | 54.8±1.6 | 67.1±0.8 | 48.6±1.4 | 53.1±1.1 | 66.9±0.9 |
| DSGD-Task | 37.3±0.6 | 45.3±0.7 | 62.4±1.2 | 43.6±1.7 | 51.1±1.3 | 57.5±2.0 | 46.7±1.2 | 51.4±0.9 | 52.7±1.6 |
| DSGD-Hybrid | 41.3±0.7 | 50.3±0.6 | 66.2±1.2 | 42.4±1.0 | 51.5±2.1 | 65.6±0.8 | 47.3±1.8 | 51.9±0.8 | 63.4±0.5 |
| Harmony | 49.3±1.2 | 51.9±1.8 | 70.9±0.5 | 45.1±1.0 | 55.3±0.7 | 67.9±1.4 | 48.2±1.1 | 52.3±0.7 | 66.2±1.6 |
| FedHGB | 51.3±0.9 | 55.0±1.3 | 69.3±0.6 | 50.0±0.9 | 54.1±1.4 | 67.9±1.6 | 48.0±1.9 | 54.3±0.6 | 66.2±0.8 |
| DMML-KD | 45.8±0.7 | 54.4±1.2 | 68.4±0.8 | 45.4±1.3 | 44.8±1.5 | 67.5±0.6 | 45.9±2.2 | 53.6±0.9 | 66.9±1.5 |
| PARSE | 51.7±1.1 | 57.3±0.8 | 71.2±0.4 | 51.9±1.5 | 57.1±1.4 | 68.4±0.8 | 49.3±1.0 | 54.6±1.3 | 68.3±1.8 |

| Agent ratios | 6 : 6 : 6 : 22 | | | | 10 : 10 : 10 : 10 | | | | 12 : 12 : 12 : 4 | | | |
|---|---|---|---|---|---|---|---|---|---|---|---|---|
| **Agent types** | **[A]** | **[V]** | **[T]** | **[AVT]** | **[A]** | **[V]** | **[T]** | **[AVT]** | **[A]** | **[V]** | **[T]** | **[AVT]** |
| DSGD-Modality | 50.4±0.9 | 50.2±1.9 | 62.1±0.6 | 70.3±1.2 | 48.9±1.2 | 50.8±1.4 | 61.9±0.5 | 71.2±0.9 | 50.1±0.5 | 52.0±1.3 | 59.7±1.0 | 71.4±0.5 |
| DSGD-Task | 50.4±1.0 | 48.7±1.5 | 57.7±0.7 | 71.3±0.6 | 51.1±0.7 | 47.0±0.9 | 58.3±0.8 | 68.1±0.9 | 51.3±0.6 | 51.3±1.4 | 60.4±1.3 | 62.5±1.2 |
| DSGD-Hybrid | 44.6±0.5 | 46.3±0.8 | 61.5±1.0 | 71.7±1.3 | 46.3±0.5 | 47.3±0.5 | 61.1±0.9 | 70.3±0.5 | 52.1±1.4 | 51.6±2.1 | 59.5±0.9 | 69.3±1.2 |
| Harmony | 50.8±1.7 | 47.6±1.5 | 61.1±1.4 | 71.6±0.7 | 51.8±0.6 | 49.1±0.7 | 59.4±1.5 | 71.2±0.5 | 52.1±0.7 | 51.5±1.1 | 59.6±1.4 | 70.0±1.2 |
| FedHGB | 50.1±1.4 | 51.2±1.8 | 62.2±0.8 | 72.2±1.0 | 49.0±1.2 | 51.0±1.2 | 61.1±0.9 | 70.5±0.7 | 53.1±0.6 | 51.9±1.3 | 60.2±0.7 | 69.6±1.2 |
| DMML-KD | 50.6±0.5 | 44.7±1.2 | 62.8±0.9 | 74.7±1.4 | 51.6±0.7 | 48.3±0.6 | 61.0±1.3 | 74.2±0.9 | 53.5±0.4 | 50.2±0.6 | 60.6±1.6 | 73.1±0.7 |
| PARSE | 53.5±0.8 | 52.3±1.4 | 64.6±1.2 | 76.2±0.9 | 53.7±0.5 | 51.4±1.2 | 63.2±0.6 | 75.1±0.7 | 54.2±1.6 | 52.1±0.4 | 61.0±1.7 | 75.2±1.2 |

● **(47%:47%:6%)**. *We select 47% of agents to miss two modalities, 47% to miss one modality, and 6% to have full modality access.*

To facilitate data allocation and ensure that each agent has access to a sufficient number of data points, we divide the dataset into 40 partitions. If an agent is assigned a specific modality configuration (e.g., [A]), the remaining modalities ([V] and [T]) are assigned to other agents. These agents are placed in different modality-based subgraphs, as described in Section 3.

Under a default non-IID setting ($\alpha = 0.5$) and a ring topology, the corresponding results are reported in Table 13. The key difference between this full combination setting and our earlier unimodal vs. all-modal comparison is that methods primarily optimized for multimodal performance—such as DSGD-Task and Harmony—struggle to capture effective cross-modal interactions, leading to degraded performance even for multimodal agents. In contrast, methods like DMML-KD and PARSE, which explicitly specialize feature learning, achieve a distinct performance profile from other methods. Notably, PARSE, our proposed approach, not only outperforms all baselines across the board, but the performance gap becomes especially pronounced for multimodal agents, highlighting its advantage in capturing synergistic and shareable representations.

# E   FEATURE SPLIT SWEEP

To examine how the feature split ratio influences performance and to assess its potential for future work, we conduct a split-sweep study on all benchmarks: KU-HAR (total dim = 192), AVE (total dim = 384), ModelNet-40 (total dim = 384), and IEMOCAP (total dim = 384). In each sweep, we vary the dimensionality of one branch while dividing the remaining budget equally between the other two. For example, setting the unique branch of AVE to $64$ dimensions assigns $160$ dimensions each to the redundant and synergistic branches, preserving the total of $384$. All other settings follow our default ablation protocol.

The Table 14 reports, on dataset KU-HAR, for each chosen split size, the overall accuracy of multimodal agents (when the split is integrated) and the split-only accuracy obtained when using

Table 13: Performance comparison on IEMOCAP across all possible agent modality combinations.

| **Agent Ratio: (30%:30%:40%)** | | | | | | |
|---|---|---|---|---|---|---|
| **Method** | [A] | [V] | [T] | [AV] | [AT] | [VT] | [AVT] |
| DSGD-Modality | 40.8±0.6 | 47.9±0.8 | 53.5±1.0 | 56.6±1.1 | 58.8±1.4 | 56.7±0.7 | 58.0±0.6 |
| DSGD-Task | 40.7±2.5 | 47.3±1.9 | 48.0±0.8 | 51.3±0.9 | 54.6±1.2 | 60.4±0.6 | 63.5±0.9 |
| DSGD-Hybrid | 38.0±1.4 | 47.1±1.5 | 52.7±1.0 | 54.1±0.8 | 60.4±1.1 | 61.7±0.7 | 64.0±0.8 |
| Harmony | 37.6±2.2 | 47.5±1.3 | 51.3±1.0 | 47.3±0.6 | 55.7±1.6 | 57.4±1.4 | 54.6±1.0 |
| FedHGB | 40.3±1.0 | 48.4±1.4 | 50.7±1.5 | 51.6±1.7 | 58.4±1.8 | 59.5±1.1 | 59.8±0.9 |
| DMML-KD | 37.1±1.0 | 48.2±1.0 | 53.3±1.1 | 60.6±0.8 | 61.9±1.1 | 63.1±1.4 | 68.8±1.3 |
| PARSE | 40.9±1.4 | 48.8±0.6 | 54.7±0.9 | 61.3±0.6 | 64.2±0.5 | 65.4±0.8 | 70.4±0.7 |

| **Agent Ratio: (43%:43%:14%)** | | | | | | |
|---|---|---|---|---|---|---|
| **Method** | [A] | [V] | [T] | [AV] | [AT] | [VT] | [AVT] |
| DSGD-Modality | 42.7±1.9 | 48.3±1.1 | 55.7±0.6 | 52.2±1.2 | 59.7±1.3 | 55.8±0.7 | 57.1±1.0 |
| DSGD-Task | 41.5±1.7 | 47.2±1.7 | 53.9±0.5 | 55.2±0.6 | 62.4±0.7 | 58.9±0.8 | 61.0±0.5 |
| DSGD-Hybrid | 42.1±1.3 | 47.8±0.6 | 56.2±0.9 | 54.3±0.5 | 62.4±0.9 | 61.0±0.5 | 62.3±0.7 |
| Harmony | 41.4±1.2 | 46.2±0.8 | 55.7±0.3 | 50.0±0.4 | 58.5±0.4 | 55.4±1.0 | 54.1±1.8 |
| FedHGB | 42.1±1.8 | 47.0±1.7 | 54.6±0.7 | 52.7±0.8 | 60.5±1.3 | 57.1±1.3 | 59.1±0.7 |
| DMML-KD | 42.7±1.8 | 47.2±0.9 | 56.1±0.8 | 58.5±1.3 | 64.7±0.5 | 62.3±0.9 | 67.5±0.7 |
| PARSE | 43.2±1.0 | 48.8±0.9 | 56.4±0.7 | 61.2±0.7 | 66.3±1.2 | 64.4±0.9 | 69.0±0.7 |

| **Agent Ratio: (47%:47%:6%)** | | | | | | |
|---|---|---|---|---|---|---|
| **Method** | [A] | [V] | [T] | [AV] | [AT] | [VT] | [AVT] |
| DSGD-Modality | 44.3±1.6 | 46.3±1.0 | 54.4±0.6 | 49.3±0.7 | 58.3±0.9 | 58.3±0.7 | 61.3±0.9 |
| DSGD-Task | 44.1±1.4 | 45.5±0.6 | 53.5±0.5 | 51.3±1.3 | 57.8±0.4 | 61.1±0.9 | 62.9±0.8 |
| DSGD-Hybrid | 44.7±0.7 | 45.3±1.1 | 54.3±1.0 | 51.3±0.6 | 58.8±0.3 | 62.0±0.4 | 66.4±0.6 |
| Harmony | 42.6±1.1 | 43.3±0.6 | 54.1±0.9 | 45.6±0.8 | 54.3±0.5 | 58.3±1.0 | 58.2±1.1 |
| FedHGB | 43.3±1.4 | 46.6±0.4 | 53.1±0.6 | 49.6±0.5 | 59.1±1.0 | 60.4±0.7 | 60.6±1.3 |
| DMML-KD | 43.1±1.2 | 45.7±0.7 | 54.7±1.1 | 57.2±0.7 | 63.3±0.7 | 63.4±1.6 | 69.0±1.5 |
| PARSE | 45.8±1.1 | 47.6±1.0 | 55.3±0.6 | 58.3±0.7 | 65.1±0.6 | 65.5±0.5 | 71.7±0.5 |

Table 14: Performance (Acc.) vs. feature split size across datasets.

(a) KU-HAR

| **Varying Split** | **Metric** | **32d** | **64d** | **96d** | **128d** |
|---|---|---|---|---|---|
| Unique | Unique-only | 83.5 | 86.2 | 86.7 | 86.1 |
|  | Combined | 87.8 | 88.6 | 88.8 | 88.5 |
| Redundant | Redundant-only | 87.2 | 87.8 | 87.4 | 87.6 |
|  | Combined | 88.7 | 88.6 | 87.6 | 87.0 |
| Synergistic | Synergistic-only | 54.5 | 61.5 | 63.7 | 67.2 |
|  | Combined | 86.9 | 88.6 | 89.5 | 88.3 |

(b) AVE

| **Varying Split** | **Metric** | **64d** | **128d** | **192d** | **256d** |
|---|---|---|---|---|---|
| Unique | Unique-only | 59.2 | 61.7 | 61.5 | 61.3 |
|  | Combined | 63.6 | 64.7 | 62.8 | 61.9 |
| Redundant | Redundant-only | 58.9 | 63.1 | 63.3 | 62.5 |
|  | Combined | 63.1 | 64.7 | 63.9 | 62.6 |
| Synergistic | Synergistic-only | 26.2 | 34.4 | 35.9 | 36.2 |
|  | Combined | 61.4 | 64.7 | 65.5 | 62.3 |

(c) ModelNet-40

| **Varying Split** | **Metric** | **64d** | **128d** | **192d** | **256d** |
|---|---|---|---|---|---|
| Unique | Unique-only | 72.3 | 75.7 | 76.2 | 76.6 |
|  | Combined | 79.2 | 79.3 | 79.7 | 80.9 |
| Redundant | Redundant-only | 74.1 | 75.3 | 76.8 | 77.2 |
|  | Combined | 79.6 | 79.3 | 81.2 | 81.4 |
| Synergistic | Synergistic-only | 48.0 | 51.5 | 56.3 | 61.8 |
|  | Combined | 80.4 | 79.3 | 79.6 | 78.6 |

(d) IEMOCAP

| **Varying Split** | **Metric** | **64d** | **128d** | **192d** | **256d** |
|---|---|---|---|---|---|
| Unique | Unique-only | 51.3 | 67.3 | 68.2 | 69.0 |
|  | Combined | 69.8 | 73.2 | 71.7 | 70.3 |
| Redundant | Redundant-only | 57.5 | 70.3 | 64.7 | 63.2 |
|  | Combined | 68.4 | 73.2 | 70.5 | 70.2 |
| Synergistic | Synergistic-only | 37.8 | 45.1 | 47.5 | 51.5 |
|  | Combined | 69.6 | 73.2 | 70.7 | 68.3 |

only that split. We observe that enlarging the unique branch from $32d$ to $128d$ lifts its stand-alone accuracy (+3 pp) but the overall accuracy plateaus (88.5–88.8 %). Oversizing redundancy even degrades overall accuracy (88.7 to 87.0 %). Synergy helps most at a moderate size (overall peak 89.5 % at $96d$). Similarly, on AVE, we observe that unique-only and redundant-only scores saturate beyond $128d$, while synergy yields the best overall accuracy at a mid-range size (65.5 % at $192d$).

Over-allocating any single branch hurts the combined model. On ModelNet-40 and IEMOCAP, the results are similar, and the overall performance is more sensitive to redundant feature size.

Balanced capacity across unique, redundant, and synergistic features—our default even split—offers a robust trade-off. Because the optimum is dataset-dependent, data-driven adaptive allocation is indeed an important avenue for future work.

# F  Fusion Method Comparison

By default, we use a simple averaging as the fusion method. To assess whether it is too crude for capturing synergistic information, we compared our default mean fusion with five stronger operators: concatenation+Linear, summation+Linear, gated fusion (Xue & Marculescu, 2023), cross attention (Zhang et al., 2022), and Hadamard product (Kim et al., 2017), where "Linear" means a linear map that can be learned. Results on all benchmarks are summarized in Table 15.

Table 15: Comparison of fusion methods across datasets (mean $\pm$ std).

(a) KU-HAR

| Fusion Method | Overall (%) | Synergy (%) |
| --- | --- | --- |
| Mean (default) | 88.6±0.5 | 61.5±1.0 |
| Concatenation + Linear | 87.1±0.9 | 60.3±0.8 |
| Summation + Linear | 89.0±1.1 | 62.2±2.1 |
| Gated Fusion | 89.0±1.4 | 64.7±1.5 |
| Cross-Attention | 89.2±1.2 | 62.1±0.7 |
| Hadamard Product | 88.6±1.3 | 58.7±1.8 |

(b) AVE

| Fusion Method | Overall (%) | Synergy (%) |
| --- | --- | --- |
| Mean (default) | 64.7±1.3 | 34.4±1.6 |
| Concatenation + Linear | 63.2±1.5 | 33.0±0.9 |
| Summation + Linear | 65.4±1.0 | 39.3±1.3 |
| Gated Fusion | 63.8±1.1 | 33.9±1.7 |
| Cross-Attention | 64.2±1.9 | 34.3±1.0 |
| Hadamard Product | 65.8±1.6 | 38.5±1.4 |

(c) ModelNet-40

| Fusion Method | Overall (%) | Synergy (%) |
| --- | --- | --- |
| Mean (default) | 79.3±0.9 | 51.5±0.8 |
| Concatenation + Linear | 79.7±1.4 | 51.9±0.4 |
| Summation + Linear | 80.8±0.6 | 53.2±1.2 |
| Gated Fusion | 81.4±0.5 | 56.1±1.0 |
| Cross-Attention | 79.6±0.8 | 51.7±0.6 |
| Hadamard Product | 80.4±0.5 | 52.7±0.3 |

(d) IEMOCAP

| Fusion Method | Overall (%) | Synergy (%) |
| --- | --- | --- |
| Mean (default) | 73.2±0.6 | 42.3±1.2 |
| Concatenation + Linear | 71.7±1.3 | 41.1±0.7 |
| Summation + Linear | 70.2±1.0 | 38.3±0.8 |
| Gated Fusion | 74.8±0.7 | 46.5±1.4 |
| Cross-Attention | 70.3±0.6 | 39.6±1.3 |
| Hadamard Product | 67.5±0.9 | 37.2±0.4 |

As shown in the results, gated fusion yields the highest synergistic-only and overall performance, improving accuracy on ModelNet-40 and IEMOCAP (by respectively $+2$ and $+1.5$ pp compared to mean fusion). Notably, PARSE performs slightly better with cross-attention on KU-HAR, while on AVE it achieves its best results with summation+Linear fusion. While some heavier operators can improve synergistic feature alignment on specific datasets, they also introduce learnable parameters that must be exchanged across agents each round. This exchange increases both bandwidth consumption and sensitivity to non-IID drift. By contrast, mean fusion minimizes coordination costs while still providing competitive accuracy, making it a practical baseline in server-free (DFL) environments.

A key advantage of PARSE is that it is *fusion-agnostic*. Our default choice is mean fusion—a lightweight, parameter-free operator that scales linearly with the number of neighbors and avoids synchronization overhead in decentralized settings. The framework itself does not depend on this choice, and any richer operators can be adopted to the framework.

# G  Feature Visualization

To provide a straightforward visualization of how PARSE captures synergistic features, we adopt the default experimental settings from Section 4. Figure 8 presents t-SNE plots of the synergistic features for each modality (before fusion) alongside the fused features. On AVE, the fused synergistic features form well-separated class clusters with larger inter-class margins and reduced intra-class scatter compared to pre-fusion features. On IEMOCAP, the fusion further separates overlapping classes, with clearer distinction between the yellow and red clusters. For ModelNet-40, where both modalities are images, the features are more aligned than complementary, reflecting their modality similarity. Overall, these results demonstrate that PARSE effectively captures complementary information, particularly in settings with heterogeneous modalities such as audio and video.

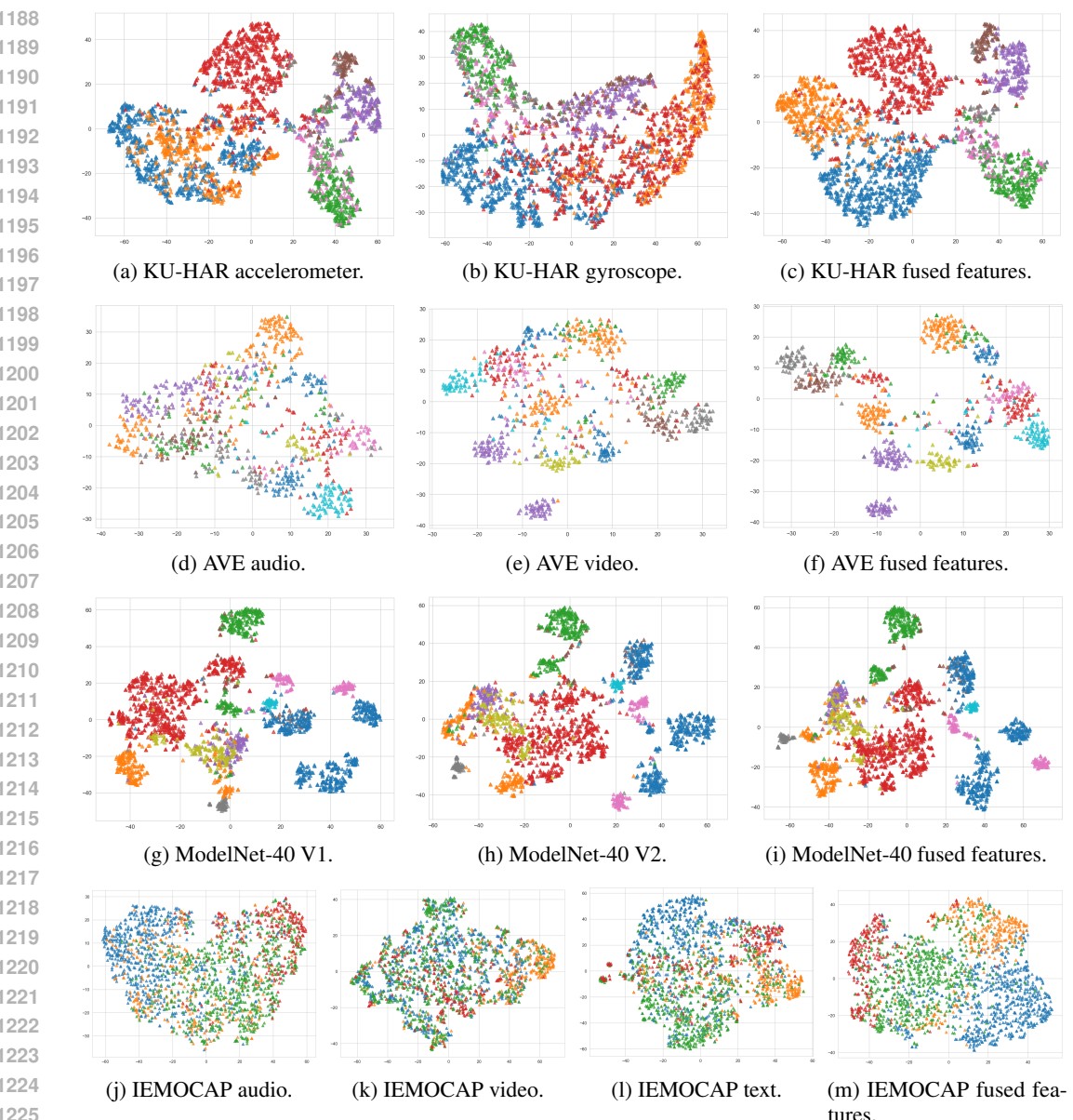

(a) KU-HAR accelerometer.  (b) KU-HAR gyroscope.  (c) KU-HAR fused features.

(d) AVE audio.  (e) AVE video.  (f) AVE fused features.

(g) ModelNet-40 V1.  (h) ModelNet-40 V2.  (i) ModelNet-40 fused features.

(j) IEMOCAP audio.  (k) IEMOCAP video.  (l) IEMOCAP text.  (m) IEMOCAP fused features.

Figure 8: t-SNE visualization of synergistic features from each modality and after fusion, showing that cross-modal fusion yields more discriminative representations for classification.

## H   FEDERATED LEARNING COMPARISON

While PARSE demonstrates strong performance in decentralized federated learning (DFL) compared to adapted FL methods under the same setting, it is also important to evaluate its effectiveness in a centralized FL scenario against server-based designs. In this section, in addition to the common baselines used in the DFL experiments, we adapt the DFL variants as follows: Fed-Modality, where each modality is trained independently and the central server aggregates encoders and classifiers within each modality group; Fed-Task, where agents perform modality-fusion learning locally and parameters are aggregated only among agents with identical modality sets; and Fed-Hybrid, which aggregates parameters by modality while still allowing local modality-fusion learning.

Beyond existing baselines, we also compare against FL methods that explicitly rely on server-side coordination: FedMSplit (Chen & Zhang, 2022), which dynamically learns inter-agent relationship

graphs to guide aggregation; FedMVD (Gao et al., 2025), which employs global alignment to mitigate domain shifts caused by modality-based heterogeneity; and FedMVC (Chen et al., 2024), a multi-view approach designed to reduce modality heterogeneity across all agents.

We adopt the same number of agents, agent ratios, and Non-IID level as in the default setting, and report the results in Table 16. As shown, even without a dedicated server-side design, PARSE achieves leading performance across all multimodal agents (notably on IEMOCAP) and most unimodal agents (with the exception of text-only agents on IEMOCAP). These findings highlight the strong extension potential of PARSE in standard FL settings, suggesting that further gains may be realized by combining it with widely used server-based FL designs, which we leave as future work.

Table 16: Comparing methods on four benchmarks in a federated learning setting. We report accuracy (ACC, higher is better) on different modalities (Non-IID Alpha=0.5).

(a) KU-HAR

| Method | A | G | AG |
|---|---|---|---|
| Fed-Modality | 85.2±0.8 | 79.1±1.3 | 91.3±0.3 |
| Fed-Task | 83.2±1.2 | 76.5±1.4 | 85.7±0.8 |
| Fed-Hybrid | 83.0±0.5 | 72.5±0.4 | 89.5±0.7 |
| Harmony | 83.7±1.3 | 78.6±0.3 | 90.5±0.6 |
| FedHGB | 82.2±1.2 | 75.4±0.5 | 88.3±0.9 |
| DMML-KD | 85.2±0.6 | 77.6±1.1 | 92.1±0.4 |
| FedMVC | 85.6±1.5 | 81.2±1.1 | 92.7±0.9 |
| FedMSplit | 82.1±0.9 | 80.7±1.1 | 90.2±1.3 |
| FedMVD | 85.1±1.4 | 78.8±1.0 | 92.5±0.8 |
| PARSE | **85.8±0.9** | **81.5±0.5** | **93.3±1.2** |

(b) AVE

| Method | A | V | AV |
|---|---|---|---|
| Fed-Modality | 53.4±1.3 | 57.3±0.9 | 72.6±1.2 |
| Fed-Task | 46.5±0.7 | 51.2±0.4 | 65.7±1.3 |
| Fed-Hybrid | 48.7±1.0 | 54.9±0.8 | 71.2±1.1 |
| Harmony | 53.1±0.5 | 56.9±0.8 | 71.6±0.7 |
| FedHGB | 52.2±1.4 | 56.4±1.5 | 70.5±0.6 |
| DMML-KD | 52.7±1.7 | 56.2±1.5 | 71.6±0.9 |
| FedMVC | 52.9±1.5 | 57.5±1.7 | 73.1±1.3 |
| FedMSplit | 47.7±1.5 | 52.9±1.3 | 70.1±1.0 |
| FedMVD | 45.2±1.0 | 54.7±1.5 | 70.4±1.5 |
| PARSE | **54.2±1.1** | **57.9±1.3** | **73.3±0.7** |

(c) ModelNet-40

| Method | V1 | V2 | V1,V2 |
|---|---|---|---|
| Fed-Modality | 85.0±1.3 | 78.3±0.8 | 86.9±0.4 |
| Fed-Task | 81.2±0.4 | 75.4±1.8 | 77.3±1.4 |
| Fed-Hybrid | 81.1±0.7 | 77.8±0.6 | 82.5±0.8 |
| Harmony | 84.6±0.5 | 81.6±1.2 | 85.6±1.6 |
| FedHGB | 81.9±1.4 | 78.1±0.5 | 82.3±1.5 |
| DMML-KD | 84.8±0.5 | 80.2±0.9 | 85.1±0.4 |
| FedMVC | 85.7±1.6 | 81.6±0.8 | 86.7±1.6 |
| FedMSplit | 81.2±1.0 | 79.3±0.8 | 87.3±1.2 |
| FedMVD | 83.6±1.2 | 78.8±1.6 | 87.4±1.3 |
| PARSE | **86.1±0.8** | **83.3±1.1** | **88.7±0.9** |

(d) IEMOCAP

| Method | A | V | T | AVT |
|---|---|---|---|---|
| Fed-Modality | 48.6±0.4 | 52.6±0.3 | **63.6±0.7** | 70.6±0.9 |
| Fed-Task | 29.6±0.6 | 45.3±0.3 | 53.6±0.5 | 67.4±0.6 |
| Fed-Hybrid | 41.3±0.3 | 52.8±0.3 | 61.0±0.7 | 69.2±0.5 |
| Harmony | 35.4±0.4 | 52.5±0.8 | 61.1±0.5 | 68.3±0.7 |
| FedHGB | 48.0±0.3 | 52.3±0.5 | 63.1±0.4 | 73.6±0.5 |
| DMML-KD | 46.0±0.4 | 53.1±0.3 | 59.2±0.5 | 74.3±0.7 |
| FedMVC | 47.1±0.8 | 52.3±0.2 | 63.2±0.6 | 73.9±1.0 |
| FedMSplit | 47.2±1.7 | 52.1±1.1 | 60.3±0.9 | 71.9±0.8 |
| FedMVD | 44.3±0.7 | 52.3±0.8 | 57.9±0.6 | 72.9±0.7 |
| PARSE | **49.6±1.5** | **55.1±1.2** | 61.7±0.5 | **77.8±0.7** |

## I  PARTIAL INFORMATION DECOMPOSITION BEYOND TWO MODALITIES

As discussed in Section 2, we extend the notion of information decomposition to handle more than two input variables, where each variable corresponds to a modality. Following the formulation in Griffith & Koch (2014), we consider $n$ random variables (modalities) $X_1, \ldots, X_n$ and a target variable $Y$; then, the total mutual information $I(X_1, \ldots, X_n; Y)$ can be expressed as a sum of modality-specific unique information, shared redundant information, and jointly emergent synergistic information:

$$I(X_1, \ldots, X_n; Y) = \underbrace{\sum_{i=1}^{n} U_{X_i}(Y)}_{\text{Unique}} + \underbrace{R(Y)}_{\text{Redundant}} + \underbrace{S(Y)}_{\text{Synergistic}}, \tag{9}$$

This decomposition breaks down the total mutual information between all variables $(X_1, \ldots, X_n)$ and the target $Y$ into:

- **Redundant Information** $R(Y)$: the information about $Y$ that is simultaneously present in all $X_i$,

- **Unique Information** $U_{X_i}(Y)$: the information about $Y$ that is available only in $X_i$ and not in the other modalities,

- **Synergistic Information** $S(Y)$: the information about $Y$ that emerges only when multiple modalities are considered together.

**Redundant information** can be further derived as

$$R(Y) = I_{\min}(\{X_1, \ldots, X_n\}; Y) \tag{10}$$

$$= \sum_{y \in \mathcal{Y}} p(y) \min_i I(X_i; Y = y), \tag{11}$$

**where:**

- $I(X_i; Y = y)$ is the pointwise mutual information between $X_i$ and the event $Y = y$.

- $I_{\min}$ approximates the minimum mutual information from any single modality alone.

- For each class label $y$, we take the minimum mutual information across all modalities—capturing the information that is commonly available across them.

- The expectation over $p(y)$ sums this effect across all possible outcomes of $Y$.

**Synergistic Information** follows the derivation:

$$S(Y) = I(X_1, \ldots, X_n; Y) - I_{\max}(\{X_1, \ldots, X_n\}; Y) \tag{12}$$

$$= I(X_1, \ldots, X_n; Y) - \sum_{y \in \mathcal{Y}} p(y) \max_i I(X_i; Y = y), \tag{13}$$

**where:**

- $I(X_1, \ldots, X_n; Y)$ is the total mutual information between all input modalities and the target.

- $I_{\max}$ approximates the best possible information available from any single modality alone.

- Subtracting this maximum single-modality information from the total information quantifies the gain achieved by combining modalities, which is the synergy.

**Unique Information** can be quantified by

$$U_{X_i}(Y) = I(X_i; Y) - I_{\min}(\{X_1, \ldots, X_n\} \setminus X_i; Y), \tag{14}$$

**where:**

- $I(X_i; Y)$ is the mutual information between modality $X_i$ and target $Y$.

- $I_{\min}(\{X_1, \ldots, X_n\} \setminus X_i; Y)$ denotes the redundant information in all other modalities excluding $X_i$.

- The subtraction gives the portion of $I(X_i; Y)$ that is *not* redundant with any other modality—i.e., uniquely contributed by $X_i$.

## J  THE USE OF LARGE LANGUAGE MODELS

During the preparation of this work, we used large language models solely to assist with polishing the writing.

