# OpenReview forum: "PID-Guided Partial Alignment for Multimodal Decentralized Federated Learning"
_ICLR.cc/2026/Conference — ICLR 2026 Conference Withdrawn Submission_

### Official Review · Reviewer_w5og · 2025-10-27

**Soundness:** 1
**Presentation:** 2
**Contribution:** 1
**Rating:** 2
**Confidence:** 5

**Summary:**

This paper proposes the PARSE framework for decentralized multimodal FL scenarios, aiming to address the challenge that unimodal and multimodal clients have inconsistent gradient directions when updating shared parameters. To address this, the paper leverages Partial Information Decomposition (PID) to split features into redundant, unique, and synergistic slices. Due to client heterogeneity, slice-level knowledge sharing is performed only when they share alignable slices. The benefit of this approach is that it can prevent interference from irrelevant parts of the model. The paper compares the proposed method with 6 baselines on 4 datasets, and the results demonstrate that the proposed method outperforms baseline methods under various settings.

**Strengths:**

This paper uses information decomposition to address modality heterogeneity among clients, which is an interesting and intuitive approach. The experimental section appears clear, including various datasets, modality heterogeneity settings, class heterogeneity settings, different topologies, ablation studies, etc.

**Weaknesses:**

1. In my opinion, the biggest flaw of the proposed method is that it is impractical, in other words, no one would use this method to train models in the real world. The reason is that the method proposed in this paper modifies the model structure rather than being a pure FL algorithm. In contrast, almost all baselines compared in the paper are FL algorithms that are general to all models. This leads to the following consequences:
    - It cannot be adapted to any currently popular models without modifying their architectures. For example, if I want to fine-tune GPT-5 while keeping its architecture unchanged, the proposed method is not applicable.

    - More critically, even if I modify GPT-5 to add the three-slice structure, I cannot simply freeze the pretrained encoder and only train the classifiers. This is because the feature splitting (from the authors' code)
      ```python
      feat = self.encoders[i](x)
      feat_m, feat_c, feat_s = torch.chunk(feat, 3, dim=1)
      ```
      is completely meaningless if the encoder is frozen. the pretrained encoder was never trained to semantically separate its output into redundant, unique, and synergistic components. Therefore, the method requires training the entire model (encoder + classifiers) from scratch with the special three-slice architecture.

    - It cannot be proved that models with the special design of redundant, unique, and synergistic slices are better than models without these special designs.

   - All experimental comparisons in the paper are: three-slice special architecture vs. standard architecture. This is not a fair comparison. FL algorithm vs. FL algorithm is acceptable, FL algorithm + special architecture vs. FL algorithm + special architecture is acceptable, but FL algorithm + special architecture vs. FL algorithm is not acceptable.

2. I believe the proposed method does not truly contribute to the DFL field, for the following reasons:
    - It does not solve the unique challenges in DFL, for example, asynchronous updates, dynamic topology, etc.

    - Many multimodal FL algorithms can be very easily adapted to DFL scenarios, such as the baselines in the paper: DSGD-Modality, DSGD-Task, and DSGD-Hybrid. I believe that at the code level, only about 20 lines of code need to be modified to adapt the CFL aggregation paradigm to the DFL neighbor averaging. Obviously, I would not call such modifications a contribution to the DFL field, and therefore the same logic applies to this paper.

    - I have to say that many experimental scenarios in the paper use the simplest and most basic ring topology. Although the paper also compares chordal ring and random gossip, these are all variants of the ring topology. This obviously cannot represent DFL. What about other topological structures, such as fully-connected structures? What about dynamically changing topologies? What about highly heterogeneous connectivity (some nodes have many connections, some have few)?

    - The paper's contributions emphasize server-free, but I do not know how to construct per-modality subgraphs in the real world without server coordination. How do clients know which neighbors have the same modality?

    - The paper's contributions emphasize topology-agnostic, but I do not know how the proposed method works under a fully-connected topology, i.e., a topology where all clients are connected regardless of modality?

**Questions:**

1. I suggest systematizing the experiments on modality heterogeneity, i.e., the agent ratios in the paper, by using a configurable parameter to represent, for example, the proportion of clients with all modalities. Additionally, why are the numbers of single-modality clients the same in all experiments? If the numbers of single-modality clients are different, for example, 5 audio clients and 20 video clients, how would the performance be, especially given that the paper constructs per-modality subgraphs?

2. I suggest testing on more topological structures, especially in cases where clients have more connections, which may better represent real-world P2P networks.

3. The paper mentions in its contributions that the proposed method is compatible with time-varying random graphs, but there are no experiments to prove this, especially in multimodal scenarios. Consider a scenario where some multimodal clients, for some reason, such as sensor damage or privacy policy changes, have some modalities no longer available. How should the proposed method handle this? Or consider, alternatively, single-modality clients who purchase new devices and collect new modality data. How should the proposed method handle this?

---

> ### Author Response · Authors · 2025-11-20
> **Response to the Weakness 1.**
>
> We thank the reviewer for the detailed comments and clarify that our method is a **representation-learning module attached to standard backbones**, rather than a change to the feedforward architecture itself. However, we strongly believe that there is a major misunderstanding here.
>
> **First**, our approach is **compatible with arbitrary network backbones**. We do not alter the feedforward connections, nor do we add or remove layers from the backbone. Instead, we only apply a linear mapping and a fixed partition to the output feature of the final backbone layer, splitting it into **redundant**, **unique**, and **synergistic** slices. This partitioning head can be placed on top of any existing model (e.g., a vision encoder or an LLM) **without redesigning its internal architecture**.
>
> **Second**, regarding the concern that feature splitting would be “meaningless” if the encoder is frozen: our method is **not designed as a scheme for LLMs themselves**, but it can naturally benefit multimodal LMs during modality-encoder training, similar to how many MLLM frameworks learn to project and align non-text modalities into a shared text-embedding space. In this sense, our method follows **standard pre-training principles**: if the representation is expected to change to encode **redundant, unique, and synergistic information**, then the corresponding encoder parameters must be updated. This is **not a limitation specific to our approach**, but a **general requirement for learning new representations**.
>
> **Third**, our experiments provide evidence that the **three-slice representation is beneficial** compared to models without such a partition. In particular, DSGD-Modality uses the same communication strategy but a non-partitioned backbone, and our method consistently outperforms it (see Table 2). This directly **isolates the effect of our representation design**.
>
> **Finally**, our comparisons are fair with respect to network capacity. All methods, including baselines, use **exactly the same backbone architectures**, differing only in the **representation learning method**. Conceptually, we compare: (i) unchanged backbones with specialized representation learning + standard communication, versus (ii) baselines that either use specialized communication with standard supervised learning or alternative multimodal representation schemes. Since the underlying backbones are identical across methods, the gains cannot be attributed to larger models or special backbone architectures, but rather to the proposed representation and training strategy.

---

> ### Author Response · Authors · 2025-11-20
> **Response to the Weakness 2.**
>
> Once again, we are afraid that there is a **major misunderstanding about our work and the DFL literature**.
>
> **First**, our work focuses on the algorithmic and **representation-learning** aspects of multimodal learning in a decentralized setting. Challenges such as asynchronous updates, dynamic topology maintenance, and connectivity management are important, but they are typically treated as **systems-level issues**. Moreover, our method can in principle be combined with existing asynchronous or topology-adaptive DFL frameworks, because it **does not assume any particular synchronization scheme or overlay construction mechanism**.
>
> **Second**, we agree that many multimodal FL algorithms can be mechanically ported from centralized FL to DFL with relatively small code changes (e.g., replacing server aggregation with neighbor averaging). However, our paper explicitly shows that such “straightforward” adaptations suffer from two fundamental limitations in multimodal DFL:
> (i) **strong gradient conflict across modality-heterogeneous agents**, and
> (ii) **limited information sharing when no mechanism explicitly discovers and leverages sharable cross-modality structure**.
> Our method is designed precisely to address these issues in a **gradient-surgery-free** and **coordination-free** manner, which goes beyond a simple change in the aggregation primitive.
>
> **Third**, the construction of overlays (ring, chordal ring, random gossip, etc.) and neighbor discovery protocols is a well-studied problem in both classical ad-hoc networks and modern P2P systems, and is largely independent of how local multimodal representations are learned. In this paper we deliberately adopt standard, widely used topologies (ring and its chordal/rand-gossip variants) to isolate the effect of the learning algorithm rather than proposing yet another topology design. Importantly, **our algorithm does not rely on any special property of the ring**; it only requires that each client can communicate with a set of neighbors, so it **naturally extends to arbitrary, even highly heterogeneous, graphs**.
>
> **Fourth**, regarding fully-connected and dynamic topologies: we already evaluate our method in a centralized FL-style setting with server-based aggregation, which is equivalent to a **fully-connected topology** where every client can aggregate with all others. We additionally report results under **random gossip, where neighbors are resampled over time**, which is a standard model of **dynamic overlays**. These experiments demonstrate that our method is not tied to a specific topology and continues to provide benefits in both fully-connected and dynamically changing settings.
>
> **Finally**, on **per-modality subgraphs and server-free operation**: in practice, clients can discover each other’s modality types through lightweight metadata exchange (e.g., advertising modality capabilities in join messages or application-level registration), without requiring a parameter server to coordinate gradient updates. Our “coordination-free” claim refers to the absence of a **central parameter server for model aggregation**; it does not preclude the existence of **minimal control-plane metadata** that allows agents to identify neighbors with compatible modalities. Under a fully-connected topology, this simply means that each client filters its (potentially large) neighbor set by modality when forming per-modality subgraphs, and our algorithm applies without modification.

---

> ### Author Response · Authors · 2025-11-20
> **Response to the Questions.**
>
> Thanks for the questions.
>
> **First**, regarding the configuration of **modality-heterogeneous populations** (e.g., ratios of single-modality vs.\ multimodality clients): in this work, we fixed the counts of single-modality clients across experiments to control for the total number of clients and to **isolate the effects of our representation-learning method**. Our **per-modality subgraph construction** is parameterized only by which clients possess a given modality, and therefore naturally generalizes to imbalanced settings such as “5 audio-only and 20 video-only clients”. In that case, the audio and video subgraphs simply have different sizes and degrees, but the learning rule and update mechanism remain unchanged.
>
> **Second**, on **topological structures**: we agree that high-degree and richer overlays are important for modeling real-world P2P networks. In our experiments, we already study several representative cases with increasing connectivity: (i) a ring (2 neighbors), (ii) a chordal ring (3 neighbors), and (iii) a centralized FL scenario, which is equivalent to a fully-connected topology where every client aggregates with all others. **Our method shows consistent gains across these settings.** That said, nothing in our algorithm is tied to these specific graph families; it only requires neighbor exchanges within per-modality subgraphs.
>
> **Third**, regarding **time-varying random graphs and changing modality availability**: we already include experiments on random gossip overlays, where each client’s neighbors are resampled over time. This corresponds to a standard model of dynamic topology and demonstrates that our method remains effective when the graph is time-varying. Scenarios like sensor failures or policy changes (leading to the loss of a modality) and device upgrades (adding a new modality) **can be handled within the same framework**: when a modality disappears for a client, the client simply stops participating in that modality’s subgraph; when a new modality becomes available, the client joins the corresponding per-modality subgraph. These behaviors are compatible with existing churn-handling and overlay-maintenance mechanisms in P2P systems, and our algorithm does not assume static graphs.

---

### Official Review · Reviewer_N2ct · 2025-10-27

**Soundness:** 3
**Presentation:** 3
**Contribution:** 2
**Rating:** 4
**Confidence:** 4

**Summary:**

This paper explores the aggregation conflict challenges faced by different client-mode architectures in decentralized federated learning and proposes a server-free multimode decentralized federated learning framework named PARSE. The framework achieves peer-to-peer knowledge sharing between heterogeneous modal frameworks by partitioning the latent features of the data into three distinct slices: redundant, unique, and synergistic. This approach effectively resolves the collaboration issues among heterogeneous modal nodes. Extensive experimental results demonstrate the effectiveness of PARSE in practical applications.

**Strengths:**

1. PARSE employs a novel approach to knowledge sharing by partitioning data features into three slices. This method is quite intriguing, as it successfully facilitates knowledge sharing and transmission through the alignment of these slices.
2. The research focuses on the issue of modal heterogeneity among agents, a relatively new field that provides significant impetus for the advancement of multimodal federated learning.

**Weaknesses:**

1. This paper employs Partial Information Decomposition to partition features into three slices. Is there a theoretical explanation supporting its effectiveness in the multimodal domain? How is the specific feature partitioning process conducted?
2. The article mentions achieving knowledge sharing through feature fission, yet the specific design involves aggregating modules from the same modality model. For example, the optimization directions for single-modal and multi-modal clients sharing the same modality differ. How is this addressed when resolving gradient conflicts?
3. The expression of the method from a peer-to-peer perspective seems somewhat odd, as it does not reflect any special design for the peer-to-peer environment. Its aggregation design is similar to other methods, aggregating parameters of the same modality. In a peer-to-peer setting, the number of neighboring nodes for each client is typically limited, and the experimental details should briefly address this setup.

**Questions:**

1. How does the PID technique specifically partition data features into three slices?
2. The article needs to provide further explanation on how gradient conflicts are resolved for single-modality or multimodality.
3. More detailed clarification is needed regarding the improvements and settings in the peer-to-peer context.

---

> ### Author Response · Authors · 2025-11-20
>
> ### **Response to Weakness 1 and Question 1.**
> Thank you for the comment. In classical information-theoretic work [1], the effectiveness of formulating multi-variable interactions within the PID framework has already been established. Our paper therefore **does not aim to re-derive these foundations**; instead, we focus on addressing key challenges in multimodal DFL and **use PID as a principled guideline** for decomposing multimodal representations and aligning sharable information between agents.
>
> The feature partition is implemented as a **dimensional partition of the final-layer latent feature vector**, as detailed in Section 3.2, Equation (3). In this process, we do not change the backbone architecture, for simplicity.
>
> [1] Virgil Griffith and Christof Koch. Quantifying synergistic mutual information. In *Guided Self-Organization: Inception*, pp. 159–190. Springer, 2014.
>
> ### **Response to Weakness 2 and Question 2.**
> Thank you for the valuable question. In our framework, we first **decompose each modality’s representation into multiple feature components** and then attach **separate classifier heads** to these components, each with its own supervision signal. Even though, at training time, the losses from these heads are merged into a single cross-entropy objective, this is fundamentally different from directly ensembling raw features into a single head. This design is related to **diversified ensemble learning**, where ensembling multiple specialized classifiers has been shown to increase tolerance to the failure or noise of individual signals, rather than causing harmful optimization shifts [2]. In our case, the decomposed heads allow knowledge to be shared (via the redundant/synergistic components) while containing modality-specific differences in the appropriate slices.
>
> Our ablation **“Ensemble Training Analysis”** further supports this: combining the decomposed heads yields **better performance and more stable training** than training each one independently, indicating that PARSE alleviates, rather than exacerbates, gradient conflicts between single-modal and multi-modal clients.
>
> [2] Shaofeng Zhang, Meng Liu, and Junchi Yan. The diversified ensemble neural network. In *Advances in Neural Information Processing Systems*, 2020.
>
> ### **Response to Weakness 3 and Question 3.**
> One of our main contributions is that **PARSE does not require any special communication mechanism** for the peer-to-peer environment. At the algorithmic level, our design actually **simplifies what typical centralized FL methods demand**: centralized coordination, global knowledge of all agents, or server-side gradient surgery—yet it still improves overall performance. Agents only exchange parameters within **per-modality subgraphs**: each client only needs to know which neighbors share a given modality, and does not need to know how many or which other modalities those neighbors possess. This makes PARSE **directly compatible with standard P2P neighbor-averaging schemes**.
>
> Regarding the limited number of neighbors in peer-to-peer settings, we already evaluate PARSE under several typical low-degree overlays: **ring topology** (2 neighbors per client), **chordal ring** (3 neighbors per client), **random gossip** (time-varying neighbors with small expected degree), and a **fully connected case via centralized FL** (see Appendix H).

---

### Official Review · Reviewer_EQQB · 2025-10-30

**Soundness:** 2
**Presentation:** 3
**Contribution:** 2
**Rating:** 2
**Confidence:** 4

**Summary:**

The paper introduces an interesting problem, multimodal decentralized federated learning in which different agents with various modalities need to collaborate without a central coordinator. To tackle this problem, the authors introduce PARSE, a new framework based on partial information decomposition (PID) theory to decompose modality-wise features into three components and perform selective alignment. The experimental results show that PARSE outperforms selected baselines in previous work.

**Strengths:**

* Good problem statement and clear motivation
* Good writing, easy to follow

**Weaknesses:**

* Novelty: The concept of modality decomposition, including PID-based variants, has been explored in prior work [1,2]. The authors should clearly articulate how PARSE advances beyond existing approaches and specify its distinctive contributions.
* Literature Review: The literature review should be broadened to encompass centralized multimodal learning methods or federated multimodal learning [1,2], not solely multimodal DFL. The authors are encouraged to discuss the challenges of applying centralized methods in distributed settings and to include additional baselines from these domains to substantiate PARSE’s robustness and design sophistication.
* Agent design: Assigning separate classifier heads to each decomposed feature is an unconventional choice. The rationale for using multiple classifiers within a single modality should be clarified, along with an explanation of how this architecture improves performance.
* Global collaboration: What is the difficulty of using DSGD with client design from federated multimodal learning ? How does PARSE handle this? The connection seems unclear to us.
* Main results: While the experiments span several benchmarks, all involve only a limited number of modalities. The current evidence is insufficient to demonstrate PARSE’s scalability as modality count increases (similar to [1])
* Ablation study: Given that each feature type has its own classifier, dropping certain features should not hinder inference. The authors should report PARSE’s performance when one or more feature types per modality are removed to assess robustness under partial feature availability.

[1] Nguyen et al., Learning Reconfigurable Representations for Multimodal Federated Learning with Missing Data, NeurIPS’25

[2] Liang et al., Quantifying & Modeling Multimodal Interactions: An Information Decomposition Framework, NeurIPS’23

**Questions:**

See Weaknesses

---

> ### Author Response · Authors · 2025-11-20
>
> ### **_Response to Weakness 1 "Novelty"_.**
> Thanks for your comment. Existing approaches apply PID-inspired decompositions in **centralized** multimodal learning or centralized FL settings, where all modalities are trained on a single server or under a server-based aggregation scheme. In contrast, **PARSE is explicitly designed for decentralized, server-free settings with modality-heterogeneous clients**. Our main goal is to decompose multimodal interactions in a way that **aligns clients with different modality subsets and mitigates cross-client gradient conflict**, a challenge that classic centralized cross-modality training and prior PID-based methods do not address.
>
> Moreover, [1] focuses on **centralized (server-based) FL** and still relies on a central server, which faces different challenges than the **fully decentralized, topology-agnostic, and modality-heterogeneous setting** that PARSE is designed to handle.
>
> Last but not least, thank you for advocating the reference [1]. However, we note that [1] was published **after the ICLR 2026 submission deadline**, so it was not available for us to consider in the original manuscript. Under the conference policy, submissions cannot be expected to cite or compare against post-deadline work. Using such a paper as a basis to question novelty risks judging the submission against information that was not available at submission time.
>
>
> ### **_Response to Weakness 2 "Literature Review"_.**
> Thank you for the suggestion. We already include a dedicated Related Work section in the appendix that covers representative multimodal representation learning and multimodal federated learning methods, including centralized approaches. We also already discuss the challenges of applying existing methods to the DFL setting in "Motivation Study", and addressing these challenges is precisely the main purpose of this paper.
>
>
> ### **_Response to Weakness 3 "Agent design"_.**
> Thank you for the question. Based on the PID framework, we **decompose the learned representation into different partitions**, where each partition encodes a distinct type of modality-relevant information. From an information-theoretic perspective, these components are treated as **independent factors**. Therefore, we assign a **separate classifier head to each partition** so that each type of information can be exploited and supervised individually.
>
>
> ### **_Response to Weakness 4 "Global collaboration"_.**
> In the *Motivation Study* section, we illustrate the limitations of directly applying DSGD to multimodal DFL, namely that it can (i) create **gradient conflict** among modality-heterogeneous clients and (ii) **limit information sharing** even between clients that share common modalities. These issues motivate the design of **PARSE**: by decomposing and explicitly extracting sharable information between modality-heterogeneous clients, PARSE **improves global collaboration** while remaining compatible with simple communication strategies such as standard DSGD-style neighbor averaging.
>
> ### **_Response to Weakness 5 "Main results"_.**
> Thank you for this comment. Our primary focus is on **modality-heterogeneous DFL** with semantically distinct modalities (e.g., audio, image, text), which is the standard setting in multimodal learning and in the baselines we compare against. In contrast, datasets such as PTBXL (12 ECG leads) and Sleep-EDF (5 physiological channels) in [1] mainly increase the number of sensor channels **within the same semantic domain**.
>
> Also, **PARSE is architecturally defined for an arbitrary number of modalities**: the decomposition and alignment mechanisms operate over a modality index set and do not rely on having exactly two or three modalities. The computational and communication overhead scales **linearly with the number of modality-specific slices**, so there is no algorithmic barrier to extending PARSE to 5, 12, or more modalities.
>
> In our current experiments, we already evaluate PARSE on multiple benchmarks with both **2- and 3-modality configurations** and diverse client-level heterogeneity patterns, which provides empirical evidence that the method behaves consistently as the modality configuration becomes more complex.
>
>
> ### **_Response to Weakness 6 "Ablation study"_.**
> In Section 4.2, **Table 3**, we already report **ablation results** where each feature component is evaluated both stand-alone and in combination with others, under different total feature dimension budgets. Since each component has its own classifier head, using only a subset of components at inference is equivalent to these “stand-alone” or “partial-combination” settings, and the reported numbers directly reflect PARSE’s behavior under **partial feature availability**. In fact, our results on unimodal clients already demonstrate **robustness** in such a regime: even though synergistic information is not available for unimodal agents, their performance is still superior to baselines.

---

### Official Review · Reviewer_7CPS · 2025-10-31

**Soundness:** 4
**Presentation:** 4
**Contribution:** 3
**Rating:** 6
**Confidence:** 4

**Summary:**

This paper introduces PARSE, a PID-guided feature decomposition and partial alignment framework for decentralized multimodal federated learning (DFL). The approach leverages partial information decomposition (PID) to factorize features into redundant, unique, and synergistic components, enabling selective peer-to-peer knowledge sharing without a central coordinator. Experiments across four public multimodal datasets demonstrate consistent performance improvements over representative baselines.

**Strengths:**

（1）Conceptual clarity and intuitiveness:
The proposed PID-based feature decomposition combined with partial alignment is conceptually simple yet elegant. It provides a clear and interpretable mechanism for handling cross-modal heterogeneity in decentralized settings.
（2）Strong presentation and experimental design:
The paper is well-written and well-structured, with comprehensive experiments and clear visualizations. The figures and tables effectively illustrate the advantages of the method, and the ablation studies offer solid insights into the model’s behavior and design choices.

**Weaknesses:**

（1）The method assumes that all agents solve the same underlying task (i.e., share the same label space). However, in many realistic multimodal decentralized scenarios, agents may work on related but distinct tasks. How would PARSE handle task heterogeneity? Would the PID-based decomposition and partial alignment still maintain consistent feature semantics across agents?
（2）The key designs—PID-based feature fission and slice-level alignment—could, in principle, also benefit centralized or server-based FL architectures. It would be valuable to clarify what aspects of PARSE are specifically tailored to the decentralized setting, beyond the lack of a coordinator. How does the framework uniquely address challenges such as gradient drift and topology variability in DFL?
（3）The notion of “synergy” is central to the method, yet its operational meaning in the reported experiments could be elaborated. In each dataset, what constitutes synergistic information in the DFL context? Which modalities contribute more to synergistic learning, and how is this reflected in the learned feature subspaces or cross-modal collaboration patterns?

**Questions:**

1)	Feature disentanglement is a well-established concept in conventional FL. The paper should clarify what is novel or specific about applying it in the decentralized FL scenario.
2)	The manuscript should specify the fusion strategy used for parameter exchange or aggregation among agents.

---

> ### Author Response · Authors · 2025-11-20
> **Response to Weakness (1)**
>
> Thank you for the insightful question. As stated in the paper, the current **PARSE** framework decomposes information *with respect to a given task*: all decomposed components (redundant, unique, synergistic) are defined as task-relevant for a shared label space. Under this assumption, the PID-based decomposition and partial alignment naturally maintain consistent feature semantics across agents.
>
> For scenarios where agents work on related but distinct tasks (e.g., segmentation vs. classification on the same domain), PARSE can be extended in at least two ways:
>
> 1. *Task decomposition / shared subtasks.* One can decompose the tasks themselves to identify common subtasks, or define a set of exported tasks that all agents can perform without sharing raw data. In this case, different agents agree to solve the same group of “doable” subtasks, and PARSE is applied with respect to this shared task set. Synergistic features can then be aligned across agents for these common subtasks.
>
> 2. *Introducing a shared auxiliary task.* Alternatively, we can introduce new tasks (e.g., self-supervised or contrastive objectives) that are shared by all clients. Agents then share and align features based on this auxiliary task, while treating their original private tasks as downstream. In this setting, redundant and unique components are available for sharing, and private heads for each agent’s specific task operate on top.
>
> In both cases, the PID-based decomposition remains meaningful, but the “task” with respect to which redundancy/uniqueness/synergy are defined must be chosen appropriately (shared subtasks or shared auxiliary task). Fully developing and evaluating PARSE under such task-heterogeneous setups is an interesting direction for future work, and we appreciate the reviewer for highlighting this extension.

---

> ### Author Response · Authors · 2025-11-20
> **Response to Weakness (2) and Question (1)**
>
> Thank you for the valuable suggestion. At the algorithmic level, **PARSE** is specifically designed to address challenges that are amplified in DFL. In the decentralized setting, we do *not* rely on a central server to (i) detect modality heterogeneity across agents, (ii) orchestrate which clients communicate, or (iii) control local optimization directions. The only requirement is that agents sharing a given modality are (possibly indirectly) connected within a per-modality subgraph; they do not need to know the full identities or modality sets of their neighbors. Simple parameter sharing along these subgraphs is sufficient for the decomposed slices to align, so that sharable knowledge is automatically propagated without any cross-agent coordination.
>
> As we illustrate in **Table 1**, compared to other decomposition-based methods such as **DMML-KD**, **MCARN**, and **FedHKD**, our method is **server-free**, **topology-agnostic**, and **gradient-surgery-free**, which makes it particularly suitable for **decentralized federated learning (DFL)**.

---

> ### Author Response · Authors · 2025-11-20
> **Response to Weakness (3) and Question (2)**
>
> Thank you for your comment. Different from some related works such as [1], *synergy* in our work refers to *new information that only emerges when two or more modalities are present simultaneously during training*, rather than simply “cross-modality training.” Under this definition, the notion of per-modality “contribution” is not directly applicable: the synergistic component is not attributable to any single modality in isolation, because it does not exist without their joint presence. As illustrated in Figure 5, each single modality fails to correctly classify certain data points until their features are fused; the correct decision boundary appears only when the modalities interact.
>
> A classic example is the XOR setting discussed in [2]: if the task is to compute the XOR of two input channels, neither channel alone is sufficient to determine the correct label, and the target cannot be recovered from any unimodal view. Only when the two channels are observed together does the necessary synergistic information become available. In this sense, the synergistic term captures genuinely *new joint information*, rather than a weighted sum of per-modality contributions.
>
> Also, we apologize for any confusion—the manuscript already specifies our fusion and aggregation strategies: Table 4 analyzes different feature fusion methods, while Section 3.4 provides a detailed description of the parameter aggregation / sharing mechanism among agents.
>
> [1] Lee, J. J., & Yoon, S. W. (2025, April). *Can One Modality Model Synergize Training of Other Modality Models?* In The Thirteenth International Conference on Learning Representations.
>
> [2] Virgil Griffith and Christof Koch. Quantifying synergistic mutual information. In *Guided Self-Organization: Inception*, pp. 159–190. Springer, 2014.

---

### Note · Authors · 2026-01-14

I have read and agree with the venue's withdrawal policy on behalf of myself and my co-authors.